# DISCOVERING KNOWLEDGE-CRITICAL SUBNETWORKS IN PRETRAINED LANGUAGE MODELS

## ABSTRACT

Pretrained language models (LMs) encode implicit representations of knowledge in their parameters. However, localizing these representations and disentangling them from each other remains an open problem. In this work, we investigate whether pretrained language models contain various *knowledge-critical* subnetworks: particular sparse computational subgraphs responsible for encoding specific knowledge the model has memorized. We propose a multi-objective differentiable weight masking scheme to discover these subnetworks and show that we can use them to precisely remove specific knowledge from models while minimizing adverse effects on the behavior of the original language model. We demonstrate our method on multiple GPT2 variants, uncovering highly sparse subnetworks (98%+) that are solely responsible for specific collections of relational knowledge. When these subnetworks are removed, the remaining network maintains most of its initial capacity (modeling language and other memorized relational knowledge) but struggles to express the removed knowledge, and suffers performance drops on examples needing this removed knowledge on downstream tasks after finetuning.

## 1 INTRODUCTION

Large-scale pretrained language models (LLMs) encode large amounts of relational knowledge (Petroni et al., 2019; Carlini et al., 2023; Liu et al., 2023), which they adapt to successfully transfer to downstream tasks (Wang et al., 2019b;a). Due to this success, considerable prior research has focused on better understanding the extent to which LLMs capture different types of knowledge that are necessary for these tasks (Liu et al., 2019; Safavi & Koutra, 2021; Da et al., 2021; Huang et al., 2022). In these works, models are prompted using natural language verbalizations of relational triplets, which associate head and tail entities. Tokens in the sequence representing entities (or relations) in the triplets are masked, and the model must infill or complete the sequence to demonstrate it encodes the knowledge expressed by the sequence (Bosselut et al., 2019; Jiang et al., 2020).

Despite the body of work in studying LLMs as knowledge bases, less work has focused on *where* and *how* this knowledge may be encoded by the models that capture it. The answer to these questions could potentially facilitate the development of more effective finetuning methods, which can be useful for rectifying factual errors made by language models, keeping models up to date with evolving knowledge, and preventing ethically undesirable behavior. Works in probing (Belinkov & Glass, 2019; Durrani et al., 2020; Antverg et al., 2022; Belinkov, 2022) and mechanistic interpretability (Geva et al., 2021; 2022b;a) discover hidden representations, neurons, and layers that are responsible for the expression of knowledge from these systems, but typically do not localize the knowledge accessing behavior at the weight-level. Recent work in model editing investigates whether specific knowledge in the model can be changed (De Cao et al., 2021; Dai et al., 2022; Hase et al., 2023b; Mitchell et al., 2022a;b; Meng et al., 2022; 2023; Hase et al., 2023a; Gupta et al., 2023; Jang et al., 2023; Chen et al., 2023). However, the goal of these methods is typically not to precisely localize the parameters responsible for encoding the knowledge, but instead to coarsely edit model parameters such that a new desired behavior (or knowledge) overwrites the model's preference for the old one.

In this work, we hypothesize that language models contain particular sparse computational subnetworks that are *responsible* for expressing specific knowledge relationships. We call these subnetworks *knowledge-critical* as they are necessary for the model's ability to express particular relational knowledge. As a result, when the knowledge-critical subnetwork is removed, the model's ability to express

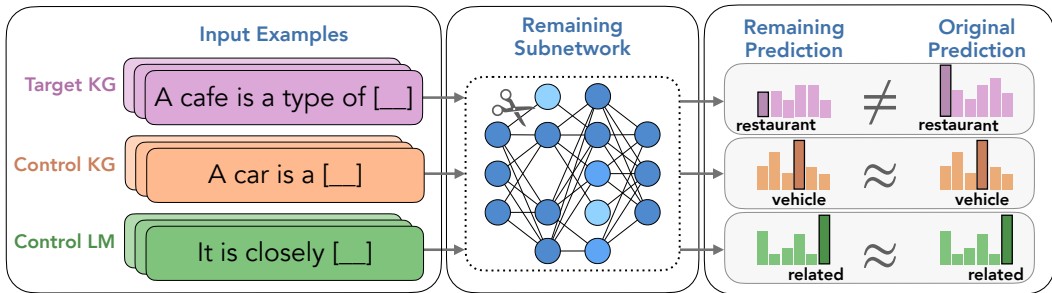

Figure 1: We hypothesize the existence of *knowledge-critical* subnetworks that are responsible for expressing target knowledge triplets (TARGETKG). When knowledge-critical subnetworks are removed, the remaining model can no longer express the specific triplets, but maintains its ability to express other relational knowledge (CONTROLKG) and its language modeling abilities (CONTROLLM). The lighter shades of blue illustrate neurons that lose weight connections in this process.

the knowledge it represents is also removed, as represented by the remaining blue model in Figure 1 that can no longer correctly predict "restaurant" as the continuation of "A cafe is a type of ___".

To discover knowledge-critical subnetworks, we propose a multi-objective differentiable weight masking method over the original pretrained model. The remaining unmasked model loses the ability to express the target knowledge on which the mask was trained, but maintains its performance on other behaviors, thereby identifying the *knowledge-critical* subnetwork as the masked portion of the original model. We combine multiple objectives designed to (1) suppress the expression of target knowledge triplets, (2) maintain an ability to express generic relational knowledge, (3) maintain standard language modeling performance, and (4) encourage the subnetwork to be as sparse as possible. Combined, these objectives optimize a mask that promotes the removal of target knowledge, while maintaining the other behaviors of the pretrained language model.

Our results — across multiple target knowledge graphs (constructed from WordNet and ConceptNet) and LLMs at multiple scales (from the family of GPT2 models) — show that our masking method consistently identifies sparse subnetworks (≈98.6% average parameters pruned) that satisfy our objectives. When these subnetworks are removed, the remaining model's perplexity on the target knowledge associated with the subnetwork largely increases (on average, a relative perplexity increase of 257% for GPT2-small, 253% for GPT2-medium, and 5589% for GPT2-large), indicating that the expression of the target knowledge is successfully suppressed. The remaining network's ability to model generic relational knowledge and natural language negligibly changes compared to the original model, implying the model maintains its original abilities. Finally, in a study on CommonsenseQA, we demonstrate that once these subnetworks are removed, models finetuned using parameter-efficient methods struggle with questions that require the knowledge encoded by the subnetwork.

## 2 RELATED WORK

**LLMs as Knowledge Bases**  Our work builds on prior research that demonstrates the memorization of large-scale language models (LLMs) pretrained on massive amounts of web data (Carlini et al., 2021; AlKhamissi et al., 2022; Carlini et al., 2023). Multiple studies have depicted the different types of knowledge encoded by LLMs, including linguistic (Liu et al., 2019; Chen & Gao, 2022), relational (Safavi & Koutra, 2021), commonsense (Da et al., 2021), and actionable knowledge (Huang et al., 2022). Parametric knowledge in LMs is typically accessed in two ways. In the first, the model is conditioned with a natural language context and must complete or infill the sequence to identify the knowledge (Petroni et al., 2019; Liu et al., 2023; Yu et al., 2023). In these studies, human-defined discrete prompts and automatic prompt engineering are used to extract single and multi-token answers from language models (Jiang et al., 2020; Shin et al., 2020; Cao et al., 2021a; Zhong et al., 2021; Qin & Eisner, 2021). Alternatively, other methods fine-tune parameters to create an interface for accessing parametric knowledge (Bosselut et al., 2019; Roberts et al., 2020; Jiang et al., 2021; Hwang et al., 2021). In contrast, our work dives deeper into *where* knowledge is encoded by LLMs and proposes an algorithm to discover the subnetworks responsible for expressing these facts.

**Function-Specific Subnetworks** Methodologically, our work draws inspiration from work that identifies task-specific subnetworks in neural networks. Perhaps most known, Frankle & Carbin (2019) proposed the *Lottery Ticket Hypothesis*, which showed that learned subnetworks could achieve test accuracy similar to that of original networks. Other works pruned subnetworks for the purpose of efficient finetuning (Mallya et al., 2018; Zhao et al., 2020; Sanh et al., 2020; Guo et al., 2021), or identifying function-specific subnetworks (Cao et al., 2021b; Sanh et al., 2020; Zhang et al., 2021; Csordás et al., 2021). Identifying function-specific subnetworks also leads to useful applications, such as disentangling representations to reduce model susceptibility to spurious correlations (Zhang et al., 2021), probing models for linguistic properties (Cao et al., 2021b; De Cao et al., 2020), and finding subnetworks specialized for different languages (Foroutan et al., 2022). Most similar to our work is that of Ren & Zhu (2022), which learned coarse subnetworks that encoded large portions of ConceptNet. Similarly to these methods, we adopt a differentiable weight masking scheme, but use it to identify highly sparse subnetworks responsible for particular expressions of knowledge.

**Mechanistic Interpretability of LLMs** Mechanistic interpretability tackles the problem of understanding model behavior by reverse-engineering computations performed by transformer models. Elhage et al. (2021) discovered algorithmic patterns and frameworks in simplified transformer models. Following this framework, researchers discovered induction heads (Olsson et al., 2022), *i.e.*, specific attention heads that can be the mechanistic source of general in-context learning in LLMs. Similarly, with interventions on multi-head self-attentions and MLP sublayers, Geva et al. (2023) identified two critical points where the model propagates information for predictions and the internal mechanism for attribute extraction. Another line of work focuses on knowledge tracing and localization in model parameters for the goal of model editing (Dai et al., 2022; Meng et al., 2022; 2023; Gupta et al., 2023; Hernandez et al., 2023). Activation patching with corrupted tokens (Meng et al., 2022) or corrupted prompts (Wang et al., 2023) use causal intervention to identify model activations responsible for flipping the model's output. In contrast, our work focuses on preserving the original model to precisely locate model *weights* responsible for expressing a given set of target knowledge without counterfactuals. Our work is closer to path patching (Goldowsky-Dill et al., 2023) and automatic circuit discovery (Conmy et al., 2023), which focus on localizing behaviors to network subgraphs, but focuses specifically on identifying subnetworks associated with knowledge relationships.

## 3 BACKGROUND

To find a knowledge-critical subnetwork in a pretrained language model, we learn a differentiable weight mask (§4) over the parameters of the LM using a knowledge prediction task where a language model is prompted for relational knowledge.

**Prompting Language Models with Knowledge Graphs** We define a global relational KG as the set of knowledge triplets $K = \{(h_1, r_1, t_1), ..., (h_k, r_k, t_k), ..., (h_n, r_n, t_n)\}$ where $h$ and $t$ are head and tail entity nodes, respectively, and $r$ is the relation that holds between the two entities. To input relational knowledge into a language model, triplets must be verbalized by instantiating a natural language template with the triplet components. For example, the knowledge triplet (house, IsA, building), can be reformulated with the IsA relation-specific template "{article} {h} is {article} {t}" as "A house is a building" A typical way to prompt for knowledge is to mask the tail entity "A house is a ___" (Petroni et al., 2019). Thus, to approximate an autoregressive model's confidence on a given triplet, we can compute a distribution over the missing token and calculate the perplexity of the actual correct token building.

**Differentiable Weight Masking for Function-Specific Parameter Search** To localize parameters that are critical for modeling specific knowledge, we learn a binary mask over each network parameter. Consider a language model $f(x, \boldsymbol{\theta})$ with pretrained parameters $\boldsymbol{\theta}$ that takes as input $x$. We learn a set of binary parameters $\boldsymbol{m} \in \{0, 1\}^{|\boldsymbol{\theta}|}$ that is element-wise multiplied with the frozen $\boldsymbol{\theta}$, such that our network is reformulated as $f(x, \boldsymbol{m} \odot \boldsymbol{\theta})$. Similar to other binary mask learning methods (Cao et al., 2021b; Sanh et al., 2020; Zhang et al., 2021), our method models each parameter mask $\boldsymbol{m}_i$ with the hard-concrete or gumbel-softmax distribution, a differentiable approach to learning continuous mask scores $\boldsymbol{s}_i \in [0, 1]$ from real-valued parameters $\boldsymbol{l}_i \in \mathbb{R}$ (Maddison et al., 2017; Jang et al., 2017):

$$\boldsymbol{s}_i = \sigma((\boldsymbol{l}_i - \log(\log \mathcal{U}_1 / \log \mathcal{U}_2))/\tau) \tag{1}$$

where $\mathcal{U}_1, \mathcal{U}_2 \sim \mathcal{U}(0, 1)$ and $\sigma$ is a sigmoid function. We use the approach of Csordás et al. (2021), which uses a straight-through estimator that thresholds the continuous score (Bengio et al., 2013):

$$\boldsymbol{m}_i = [\mathbb{1}_{\boldsymbol{s}_i > 0.5} - \boldsymbol{s}_i]_{\text{detach}} + \boldsymbol{s}_i \tag{2}$$

where $\mathbb{1}$ is an indicator function that thresholds the scores at 0.5 and $[]_{\text{detach}}$ is an operation that prevents back-propagation. This way, we back-propagate through the non-detached continuous mask scores $\boldsymbol{s}_i$ and still calculate loss with the overall binarized mask score $\boldsymbol{m}_i$.

## 4 METHODOLOGY

This section defines our methodology for finding *knowledge-critical* subnetworks with differentiable weight masking. We define our criteria for such a subnetwork and propose objectives that can optimize for the criteria.

**Notation** We define a subnetwork as in §3: $f(x, \boldsymbol{m} \odot \boldsymbol{\theta})$, where $\boldsymbol{\theta}$ is the set of parameters of the network $f$ and $\boldsymbol{m}$ is the mask over a portion of that network's parameters. We assume a target set of knowledge $K_T \subset K$ (TARGETKG) for which we want to identify the responsible parameters.

### 4.1 KNOWLEDGE-CRITICAL SUBNETWORKS

Our overall goal is to find *knowledge-critical* subnetworks, which are essential parameters to express a given set of target knowledge. When knowledge-critical subnetworks are removed, the expression of the target triplets should be suppressed, and the expression of irrelevant triplets should be unaffected.

**Suppression** For $f(x, \boldsymbol{m} \odot \boldsymbol{\theta})$ to be *critical* in expressing $K_T$, its removal from the original network should also remove the model's ability to express the knowledge in $K_T$. More formally, the inversely masked subnetwork (*i.e.*, remaining model), $f(x, \tilde{\boldsymbol{m}} \odot \boldsymbol{\theta})$, where $\tilde{\boldsymbol{m}} = 1 - \boldsymbol{m}$, should have difficulty expressing $K_T$. We define this as the **suppression** criterion, as it encourages that the remaining model cannot represent knowledge in $K_T$. If we find such a disentanglement, we consider that the pretrained model heavily relied on the removed subnetwork to perform a task related to $K_T$.

**Maintenance** However, only optimizing for **suppression** leaves the possibility that our method may discover subnetworks that are *critical* to all expressions of knowledge, or expressions of any coherent sequence of language. As the model should retain most of its initial capacities, we also define **maintenance** criteria that *knowledge-critical* subnetworks must follow: (1) they should not affect the model's original performance on other relational knowledge $K_C = K \setminus K_T$ (CONTROLKG), and (2) they should not affect the model's original language modeling abilities on a standard dataset $D_{LM}$ (CONTROLLM). We refer to these criteria as **maintenance-KG** and **maintenance-LM** respectively.

**Sparsity** Finally, we would like the percentage of parameters pruned for the critical subnetwork to be as high as possible to find the parameters that primarily encode the expression of $K_T$. There may be irrelevant parameters that are not essential to the expression of $K_T$ or $K_C$ that do not get pruned from the critical subnetwork if we do not enforce a high sparsity.

### 4.2 MASK LEARNING

To learn a weight mask for knowledge-critical subnetworks, we define a joint objective that optimizes for the criteria defined above.

**Suppression Loss** To fulfill the **suppression** criterion, the remaining model, denoted as $f(x, \tilde{\boldsymbol{m}} \odot \boldsymbol{\theta})$, should be less confident in the expression of knowledge in $K_T$. We propose to minimize the KL divergence between the remaining model's predicted distribution over possible tail entities of a knowledge triplet and a uniform reference distribution $p_u$ over the tokens in the model's vocabulary. Thus, for $x \in K_T$:

$$\mathcal{L}_{\text{suppress}} = D_{\text{KL}}(p_u \parallel f(x, \tilde{\boldsymbol{m}} \odot \boldsymbol{\theta})) \tag{3}$$

**Maintenance Losses** As there are multiple ways a model could learn to suppress the expression $K_T$, mainly (1) suppressing all knowledge that is in the same format or (2) suppressing all language expressions completely, we define two regularization objectives. To encourage the rest of the model to keep its original performance on the control knowledge $K_C$ and a standard language modeling

dataset $D_{LM}$, we calculate the KL divergence of $f(x, \tilde{\boldsymbol{m}} \odot \boldsymbol{\theta})$ with the pretrained model's distribution $f(x, \boldsymbol{\theta})$ as the reference. Therefore, for $x \in K_C$ and $x \in D_{LM}$ :

$$\mathcal{L}_{\text{maintain}} = D_{\text{KL}}(f(x, \boldsymbol{\theta}) \parallel f(x, \tilde{\boldsymbol{m}} \odot \boldsymbol{\theta})) \tag{4}$$

We define two such loss terms, one for each of **maintenance-KG** and **maintenance-LM**.

**Sparsity Regularization**  To encourage our subnetwork to be sparse for maintenance reasons (*i.e.*, reducing side effects to pretrained model behavior when removed) and so that they do not contain non-critical parameters for modeling TARGETKG (*e.g.*, redundant language modeling parameters), we minimize the average subnetwork density (*i.e.*, sigmoid of the masking parameters $\boldsymbol{l}_i$ from Eq. 1):

$$\mathcal{L}_{\text{sparsity}} = \frac{1}{|\boldsymbol{\theta}|} \sum_{i=1}^{|\boldsymbol{\theta}|} \sigma(\boldsymbol{l}_i) \tag{5}$$

**Final Loss**  Our final loss is a mixture of these losses with weights $\lambda_i$ (listed in Appendix B):

$$\mathcal{L}_{\text{final}} = \lambda_1 \mathcal{L}_{\text{suppress}} + \lambda_2 \mathcal{L}_{\text{maintain-KG}} + \lambda_3 \mathcal{L}_{\text{maintain-LM}} + \lambda_4 \mathcal{L}_{\text{sparsity}} \tag{6}$$

## 5 EXPERIMENTAL SETUP

**Models & Training**  To test whether our method can scale to various model sizes, we discover knowledge subnetwork masks for GPT2-small (117M parameters, 12 layers), GPT2-medium (345M parameters, 24 layers), and GPT2-large (774M parameters, 36 layers; Radford et al., 2019). During mask learning, we do not mask the embedding, language modeling head, layer-normalization, and bias parameters.[1] We also only learn masks for the top 50% of the transformer layers.[2] For more information on implementation and checkpoint selection, please refer to Appendix B.

**Datasets**  To create TARGETKG and CONTROLKGs, we sample hypernym triplets from WordNet (Miller, 1995), as well as triplets from the LAMA subset of ConceptNet (Speer et al., 2017; Petroni et al., 2019). For simplicity, we only consider triplets with single-token tail entities. To gather small connected TARGETKG graphs, we randomly select an initial node and sample knowledge triplets by walking a depth of three up (parent direction) and down (child direction) in the respective KG. We sample 7 TARGETKGs for WordNet using this method, and 3 for ConceptNet (statistics shown in Table 5 of the Appendix). To create CONTROLKG, we prioritize not leaking TARGETKG counterfactuals and having a shared CONTROLKG across different TARGETKGs, and so remove from the complete KG any triplet that shares the same entities as the union of the TARGETKGs shown in Table 5. For all KG verbalizations, to remove and maintain knowledge that the model is already confident about, we pick the best scoring verbalization for each triplet among several prompt styles. Statistics on TARGETKG and CONTROLKG datasets can be seen in Table 5. For the CONTROLLM dataset, we use WikiText-2 (Merity et al., 2017). We refer to CONTROLKG and CONTROLLM together as maintenance datasets. The CONTROLKG and CONTROLLM results are on the held-out validation set. Please refer to Appendix A and B for more data processing details and examples.

**Success Metrics**  Considering perplexity (PPL) as a proxy for a model's confidence in the expression of knowledge, we can reformulate the knowledge-critical subnetwork goals as:

1. **Suppression**: $\text{PPL}(f(x, \tilde{\boldsymbol{m}} \odot \boldsymbol{\theta})) \ll \text{PPL}(f(x, \boldsymbol{\theta}))$, for $x \in K_T$
2. **Maintenance-KG**: $\text{PPL}(f(x, \tilde{\boldsymbol{m}} \odot \boldsymbol{\theta})) \approx \text{PPL}(f(x, \boldsymbol{\theta}))$, for $x \in K_C$
3. **Maintenance-LM**: $\text{PPL}(f(x, \tilde{\boldsymbol{m}} \odot \boldsymbol{\theta})) \approx \text{PPL}(f(x, \boldsymbol{\theta}))$, for $x \in D_{LM}$
4. **Sparsity**: $0 < \sum_{i=1}^{|\boldsymbol{\theta}|} \boldsymbol{m}_i \ll |\boldsymbol{\theta}|$

To measure these conditions, we calculate the perplexity difference between the remaining and original models. We refer to the perplexity difference as $\Delta \, \text{PPL} = \text{PPL}(f(x, \tilde{\boldsymbol{m}} \odot \boldsymbol{\theta})) - \text{PPL}(f(x, \boldsymbol{\theta}))$.

---

[1]Prior work has not observed an advantage to masking these components for general tasks (Zhao et al., 2020).

[2]Multiple layer-wise model analyses have shown that the first few layers of transformer language models encode low-level linguistic tasks and features that may be a prerequisite for knowledge modeling (Tenney et al., 2019; Liu et al., 2019). We also perform a masked layer choice study that confirms this intuition (Appendix C).

Table 1: **Subnetwork discovery results for GPT-2 small,** averaged over three seeds with [min, max] values denoted in brackets. $\Delta$ PPL = PPL($f(x, \tilde{m} \odot \theta)$) - PPL($f(x, \theta)$). The arrows ($\uparrow$,$\downarrow$) show the desired value for the metric. Random is an average of randomly masked baselines at the same sparsity levels as the discovered knowledge-critical subnetworks for each KG-seed pair.

| Knowledge Graph | | Sparsity ($\uparrow$) | TARGETKG $\Delta$ PPL($\uparrow$) | CONTROLKG $\Delta$ PPL ($\downarrow$) | CONTROLLM $\Delta$ PPL ($\downarrow$) |
|---|---|---|---|---|---|
| WordNet | building | 98.4 [97.4, 99.3] | 62.3 [13.2, 114.1] | -2.0 [-7.0, 2.4] | 0.6 [0.3, 1.0] |
| | communication | 99.2 [99.0, 99.3] | 104.8 [61.1, 165.9] | -1.2 [-2.2, 0.0] | 0.3 [0.3, 0.3] |
| | change | 98.4 [98.0, 99.1] | 567.2 [38.7, 1405.6] | 0.6 [-1.6, 3.0] | 0.7 [0.4, 0.9] |
| | statement | 98.2 [96.3, 99.2] | 152.5 [53.5, 248.7] | -0.5 [-3.2, 2.8] | 0.8 [0.3, 1.8] |
| | location | 99.0 [98.8, 99.1] | 810.5 [469.2, 1200.7] | 0.5 [-1.7, 3.9] | 0.3 [0.3, 0.4] |
| | representation | 98.1 [97.1, 98.8] | 221.8 [115.5, 334.4] | 2.9 [0.6, 4.0] | 0.6 [0.4, 1.0] |
| | magnitude | 99.0 [98.6, 99.3] | 2216.9 [1730.7, 2665.1] | -1.8 [-2.6, -0.9] | 0.3 [0.2, 0.4] |
| | Random | 98.6 [98.1, 99.2] | 24.3 [5.0, 48.8] | 14.6 [0.0, 46.2] | 2.2 [1.2, 3.3] |
| | Average | 98.6 [98.1,99.2] | 590.9 [62.3,2216.9] | -0.2 [-2,2.9] | 0.5 [0.0, 0.8] |
| ConceptNet | fruit | 99.2 [99.1, 99.4] | 743.9 [300.8, 1462.1] | 3.0 [-0.6, 5.0] | 0.2 [0.2, 0.2] |
| | sun | 99.2 [99.0, 99.3] | 888.4 [521.0, 1240.1] | 3.2 [2.0, 4.7] | 0.2 [0.1, 0.3] |
| | swimming | 99.0 [98.8, 99.2] | 276.8 [240.9, 335.4] | 2.3 [0.6, 3.3] | 0.3 [0.2, 0.4] |
| | Random | 99.1 [99.0, 99.2] | 21.0 [13.7, 29.4] | 14.6 [12.4, 17.2] | 1.5 [1.3, 1.7] |
| | Average | 99.1 [99.0, 99.2] | 636.4 [276.8, 888.4] | 2.8 [2.3, 3.2] | 0.2 [0.2, 0.3] |

We also report the tail token rank difference between the remaining and original models in Appendix D. For the **suppression** and **maintenance-KG** criteria, we calculate $\Delta$ PPL using the loss on the masked tail entity for examples in the TARGETKG and CONTROLKG datasets. For a knowledge-critical subnetwork, we expect $\Delta$ PPL to be high for TARGETKG and low for CONTROLKG. For the **maintenance-LM** criterion, we calculate $\Delta$ PPL as the average perplexity on all tokens in a sequence, which should be low if the knowledge-critical subnetwork mask does not affect the model's general language modeling ability. For the **sparsity** criterion, we calculate the percentage of mask parameters that are 0 after the straight-through threshold in Equation 2. The denominator is the number of maskable parameters. Ideally, the sparsity should be as large as possible (*e.g.*, near 99%).

**Baseline**   As a control baseline, we create randomly masked models at the same sparsity level as the knowledge-critical subnetwork. If the discovered subnetwork is critical for expressing TARGETKG, then removing a random subnetwork at the same sparsity should have significantly lower corruption for expressing TARGETKG (*i.e.*, lower $\Delta$ PPL) than removing the critical subnetwork. Similarly, if the critical subnetwork successfully preserves the **maintenance** criteria, a random subnetwork should be more likely to remove useful weights for expressing CONTROLKG and CONTROLLM, which should lead to a higher $\Delta$ PPL on maintenance datasets. For more information on the baseline implementation, please refer to Appendix B.

## 6   EXPERIMENTAL RESULTS

### 6.1   KNOWLEDGE-CRITICAL SUBNETWORK DISCOVERY

We first evaluate the degree to which the discovered subnetworks are knowledge-critical. In Table 1, we observe that across seven different knowledge graphs (TARGETKGs) and three random seeds, the subnetworks consistently achieve a notably high sparsity ($> 98\%$) fulfilling the **sparsity** criterion, with the highest 99.3% sparsity on `magnitude` and `building` in WordNet and `fruit` in ConceptNet. For the **suppression** criterion, we notice a high $\Delta$ PPL on TARGETKG, meaning that the perplexity of the remaining model on TARGETKG is significantly higher than the pretrained model's perplexity. Specifically, the perplexity of the remaining model increases on average by around 590 for WordNet-derived TARGETKGs and 636 for ConceptNet-derived TARGETKGs compared to the original model. In contrast, removing a random subnetwork at the same sparsity only increases the perplexity on average by 24.3 for WordNet and 21 for ConceptNet, meaning the discovered subnetworks are significantly more critical for expressing TARGETKG. At the same time, we find little change in perplexity on the maintenance datasets for relational knowledge (CONTROLKG) and language modeling (CONTROLLM), demonstrated by the negligible $\Delta$ PPL on both datasets. We note that a

Table 2: **Subnetwork discovery results on GPT2-medium and GPT2-large,** averaged over two random seeds and three KGs. Random is an average of randomly masked baselines at the same sparsity levels as the discovered knowledge-critical subnetworks for each KG-seed pair.

| Model Size | | Sparsity (↑) | TARGETKG Δ PPL (↑) | CONTROLKG Δ PPL (↓) | CONTROLLM Δ PPL (↓) |
|---|---|---|---|---|---|
| Medium | Random | 96.4 [94.8, 99.5] | 32.1 [5.0, 55.6] | 9.2 [1.8, 15.9] | 3.0 [0.3, 4.9] |
| | Average | 96.4 [94.8, 99.5] | 255.6 [139.9, 432.2] | 2.5 [-0.1, 4.0] | 0.7 [0.1 , 1.2] |
| Large | Random | 98.2 [95.9, 99.6] | 6.8 [4.8, 7.8] | 2.9 [0.7, 7.3] | 0.8 [0.2, 2.1] |
| | Average | 98.2 [95.9, 99.6] | 5779.9 [1963.1, 13363.6] | 3.2 [0.9, 6.8] | 0.2 [0.0, 0.6] |

Table 3: **Ablation study for the multi-objective loss,** averaged across three KGs and two seeds.

| Ablation | Sparsity (↑) | TARGETKG Δ PPL (↑) | CONTROLKG Δ PPL (↓) | CONTROLLM Δ PPL (↓) |
|---|---|---|---|---|
| No Suppression | 99.5 [99.5, 99.5] | -7.2 [-11.9, -3.7] | -3.2 [-3.2, -3.2] | 0.2 [0.2, 0.2] |
| No Maintenance-LM | 99.2 [99.0, 99.3] | 259.8 [-1.5, 401.7] | 9.0 [-3.6, 25.1] | 25.9 [24.7, 27.3] |
| No Maintenance-KG | 99.8 [99.8, 99.8] | 21141.1 [16885.9, 25471.8] | 1697.5 [1334.6, 2180.1] | 0.2 [0.2, 0.2] |
| Our Method | 98.6 [97.8, 99.1] | 378.1 [74.3, 834.9] | 1.6 [-0.7, 4.0] | 0.5 [0.3, 0.8] |

negative Δ PPL here may result from the remaining model slightly overfitting to the CONTROLKG distribution, although it is never too significant.

**Model Scale** In Table 2, we show similar results as we scale up the original model's size and discover sparse subnetworks for larger model variants, GPT2-medium and GPT2-large. We observe an average increase in TARGETKG perplexity of 256 for GPT2-medium and 5780 for GPT2-large on the TARGETKGs, and a negligible Δ PPL on the maintenance datasets. Interestingly, we find that for GPT2-medium, our method generally finds less sparse subnetworks compared to the other model scales (∼96% sparsity). Nevertheless, the discovered subnetworks are still significantly more effective than removing an equally sparse random subnetwork (see Table 11 and Table 13 in Appendix for individual KG results).

**Ablation Study** As our method relies on a joint objective combining multiple loss functions, we perform an ablation study of the loss terms presented in §4.2, and remove each objective (i.e., No Suppression, No Maintenance-KG, No Maintenance-LM) to validate whether these losses accomplish their goals.[3] For this experiment, we focus on three TARGETKGs: communication, representation, and location, which span various degrees of suppression difficulty per the results in Table 1 (excluding the least and most suppressed TARGETKGs, building and magnitude). In Table 3, we observe that the **suppression** loss is necessary to increase TARGETKG perplexity (and remove the knowledge). Without it, the model only optimizes for retaining CONTROLKG, and generalizes this improvement to TARGETKG as well (as indicated by the negative TARGETKG Δ PPL). We also find that removing the maintenance losses directly affects CONTROLKG and CONTROLLM perplexity differences, indicating that, without these controls, our algorithm learns to remove the knowledge from the model by suppressing *general abilities*. The **suppression** objective, a minimization of the KL divergence between the output distribution and a uniform distribution, affects the prediction of tail entities for all relational knowledge rather than affecting only TARGETKG. We evaluate alternative objectives in Appendix G and do not find that they are particularly better on the success metrics than the final loss in Equation 6.

## 6.2 KNOWLEDGE-CRITICAL SUBNETWORK STRUCTURE AND COMPOSITION

In the previous section, we concluded that it is possible to find knowledge-critical subnetworks that successfully suppress TARGETKG and maintain prior abilities of the pretrained language model. Two questions that naturally arise from this success are how these subnetworks are structured and whether we can compose them (1) across random seeds for the same TARGETKG to increase the suppression effect, or (2) across TARGETKGs to suppress the union of all target knowledge simultaneously. Towards these questions, we analyze the overall structure of the knowledge-critical subnetwork. In

---

[3]We do not ablate the Sparsity term. Without it, the subnetwork search stagnates at the initial sparsity.

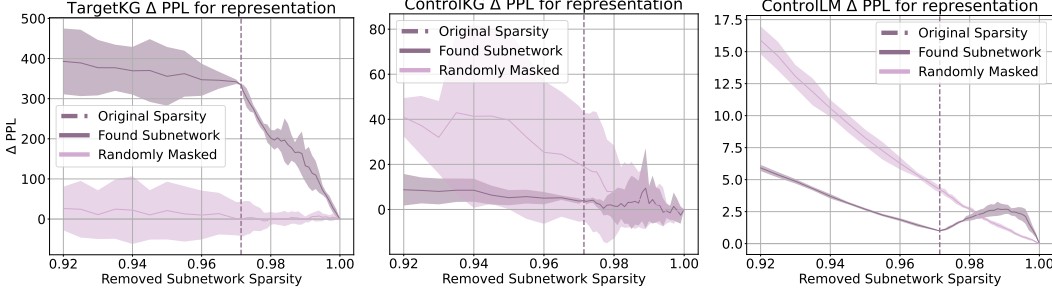

Figure 2: **Removing and adding parameters to the remaining model,** averaged over five seeds, with standard deviation depicted as the filled area around the average curves. The $x$-axis is the removed subnetwork sparsity. The $y$-axis is the $\Delta$ PPL = PPL($f(x, \tilde{\boldsymbol{m}} \odot \boldsymbol{\theta})$) - PPL($f(x, \boldsymbol{\theta})$) for the different datasets. Vertical dashed lines show the original sparsity of the critical subnetwork. The darker curve is the outcome starting from the critical subnetwork, whereas the lighter curve is from a randomly masked model at the same sparsity.

particular, we (1) analyze subnetwork density across layer depths and types, (2) calculate the Jaccard similarity (*i.e.*, Intersection-over-Union or IoU) across random seeds for the same KG and across KGs for the same random seed, and (3) evaluate the effect of naively composing subnetworks (*i.e.*, union or intersection) across seeds for the same KG and across KGs for the same seed.

We find that subnetwork masks are the densest in the first and final masked layers, particularly in attention sublayers (Figure 8). Interestingly, particular attention heads in these layer depths seem more dense across different KGs and random seeds, as shown in Figure 9. In middle layers, on the other hand, feed-forward networks are more dense. Despite this finding, the IoU of the found subnetworks across random seeds for the same KG and across KGs for the same random seed is quite low on average (3-4%) and a bit higher for the final attention output sublayer (10-12%), meaning the discovered subnetworks do not intersect as much at the weight-level. Finally, when we compose subnetworks as a union of three seed masks for the same KG or three KG masks for the same seed, we find that the suppression effect increases significantly (from an average $\Delta$PPL of 300 to $\sim$2000), although the maintenance criteria are less ideal than using an individual subnetwork (e.g. $\sim$30-40 $\Delta$PPL on CONTROLKG instead of near 0). More details can be found in Appendix I, J, and K.

### 6.3 ARE DISCOVERED SUBNETWORKS SPURIOUS SUPPRESSION SOLUTIONS?

We hypothesize that a spurious subnetwork would cause the remaining network from which it was removed to re-gain the ability to express TARGETKG if the subnetwork was randomly *expanded* (*i.e.*, $\Delta$ PPL on TARGETKG would drop as more parameters are removed from $f(x, \tilde{\boldsymbol{m}} \odot \boldsymbol{\theta})$). Meanwhile, if removing the critical subnetwork is not a spurious solution to suppress the TARGETKG, then the remaining model would generally still fail to recognize TARGETKG, even as more parameters were randomly removed, leading $\Delta$ PPL to rise or stay the same. To verify this hypothesis, we remove further parameters from the remaining model. Starting from the knowledge-critical subnetwork sparsity, we randomly remove parameters at intervals of 0.5%. We run this iterative process of removing parameters with five different random seeds. We also test whether the mask has found a spurious solution to achieve the maintenance criteria by adding back parameters, though with smaller intervals of 0.1%, as the starting sparsity level is typically high.

In Figure 2, we observe that removing more parameters in small amounts does not significantly recover expressing TARGETKG.[4] As a baseline, we plot the effect on $\Delta$ PPL of removing further parameters from remaining models with randomly removed subnetworks of the same sparsity. Interestingly, for the maintenance datasets, $\Delta$ PPL for both datasets increases as we remove parameters from the remaining model. When we add back parameters, we do not see a linear recovery to $\Delta$ PPL $= 0$. Instead, we observe an initial phase of increase followed by a phase of decrease as the model returns to its original state (*i.e.*, a $\Delta$ PPL of zero at 100% sparsity). This effect can be explained by the fact that our subnetwork had been optimized to keep these abilities, and has been slightly overfit for maintenance, though not for *suppression*. Thus, randomly adding parameters back yields new sub-

---

[4]Further plots are available in Figure 3 in Appendix E.

optimal pathways that corrupt the model's original distribution. Additional experiments on robustness, particularly to paraphrases of TARGETKG and CONTROLKG, can be found in Appendix H.

## 6.4 DO KNOWLEDGE-CRITICAL SUBNETWORKS AFFECT DOWNSTREAM TASK TRANSFER?

In our final experiment, we hypothesize that if a subnetwork is truly knowledge-critical, its removal should harm a pretrained language model's ability to transfer to a downstream task requiring the knowledge encoded by the subnetwork. To test this hypothesis, we finetune a remaining model on the challenging CommonsenseQA benchmark (Talmor et al., 2019) after removing a relevant knowledge-critical subnetwork. We use the in-house splits from Lin et al. (2019), with a development set of 1241 questions, and an initial test set of 1221. For each question in the test set, we induce[5] the ConceptNet relation associated with it (see Talmor et al., 2019 for details on data construction), and extract the facts from ConceptNet associated to this question through this relation type. Using

Table 4: **Accuracy on downstream CommonsenseQA task,** averaged over three seeds. Ours refers to removing the knowledge-critical subnetwork. Random refers to removing a random subnetwork at same sparsity as the critical subnetwork.

| Method | Subnetwork | Dev | Test | Filtered |
|---|---|---|---|---|
| **Head Tuning** | Full | 38.63 | 38.33 | 37.19 |
| | Random | -0.47 | -1.61 | -3.21 |
| | Ours | -1.69 | -6.80 | -14.42 |
| **LoRA** | Full | 50.04 | 48.64 | 48.67 |
| | Random | -0.74 | -2.33 | -1.75 |
| | Ours | -1.83 | -2.74 | -3.95 |
| **Full Finetuning** | Full | 44.61 | 42.33 | 42.79 |
| | Random | +0.30 | -0.24 | +2.39 |
| | Ours | -1.50 | -5.14 | -3.60 |

this process, we create a TARGETKG from all ConceptNet facts associated to the test set and filter non-single token relations (Filtered), yielding a filtered set of 363 questions for which we can reliably extract relevant ConceptNet triplets. For these remaining questions, we use these relevant triplets as TARGETKG, and the remaining distinct triplets in the LAMA subset of ConceptNet as CONTROLKG to learn a knowledge-critical subnetwork mask. Then, we apply different finetuning methods on the remaining model after removing the critical subnetwork, using the same training set Talmor et al. (2019). We compare to the performance of finetuning the full pretrained model (Full), as well as a randomly masked model at the same sparsity as the critical subnetwork (Random). We carry out all of these steps for three seeds and report the average accuracies in Table 4.

For all finetuning methods, we find that the remaining model has a similar accuracy as the pretrained model on the development split and a close accuracy for the overall test set. However, we observe a consistent performance drop on the filtered subset after finetuning (average performance drop of 7.3%; head tuning barely better than selecting a random answer on a 5-choice MCQA task), indicating the model does not as reliably transfer knowledge from TARGETKG during finetuning. For both head tuning and LoRA (Hu et al., 2022), we also find that if we randomly split the filtered TARGETKG set in two, one half's knowledge-critical mask does not affect the accuracy of the other half as significantly as its own (see Appendix F for more details), indicating that the performance drop is indeed specific to the knowledge that is pruned.

## 7 CONCLUSION

In this paper, we conceptualize knowledge-critical subnetworks, sparse computational subgraphs within a larger language model that are responsible for expressing specific knowledge relationships. We discover these subnetworks within the computation graphs of language models using a multi-objective differentiable weight masking approach that jointly optimizes (1) a suppression criterion designed to suppress the expression of target knowledge when knowledge-critical subnetworks are removed from a language model, and (2) multiple maintenance criteria that ensure the language model retains its ability to model other relational knowledge and general language. Our results demonstrate that when these discovered knowledge-critical subnetworks are removed, a model loses its capacity to express the knowledge encoded in the subnetwork, as well as its transfer capacity when finetuned on downstream tasks requiring the knowledge from the subnetwork.

---

[5]We describe this process in Appendix F.

REPRODUCIBILITY

For knowledge graph and language modeling datasets, we describe our sources, the creation process, and the processing and filtering steps in the "Datasets" paragraph in §5 and Appendix A. We also report how we split and process the downstream CommonsenseQA task data in §6.4 and Appendix F. Information on mask implementation and training, including details about hyperparameters, dataloaders, mask implementation, randomly masked baseline implementation, checkpoint selection, and hardware, can be found in the "Models & Training" paragraph in §5 and Appendix B. We will share the code upon publication.

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

Table 5: **Statistics on sampled KGs and their verbalization.** The graph statistics show the amount of triplets and the unique amount of heads, tails, and relations. The average perplexity is calculated with the gold tail token cross-entropy loss. The perplexity for certain KGs in the Medium and Large model columns is missing as we do not evaluate on them in our study on scale.

| Knowledge Graph | | # triplets | # heads | # tails | # rels | GPT-2 PPL | | |
|---|---|---|---|---|---|---|---|---|
| | | | | | | Small | Med | Large |
| WordNet | CONTROLKG train | 9751 | 9707 | 2709 | 1 | 63.6 | 32.8 | 27.4 |
| | CONTROLKG val. | 50 | 50 | 50 | 1 | 73.2 | 37.5 | 31.3 |
| | building | 11 | 11 | 11 | 1 | 51.9 | - | - |
| | communication | 16 | 16 | 9 | 1 | 96.3 | 65.2 | 69.2 |
| | change | 13 | 13 | 13 | 1 | 109.7 | - | - |
| | statement | 16 | 16 | 16 | 1 | 170.2 | - | - |
| | location | 19 | 19 | 7 | 1 | 198.0 | 119.0 | 125.5 |
| | representation | 12 | 12 | 12 | 1 | 210.7 | 106.8 | 108.7 |
| | magnitude | 12 | 12 | 7 | 1 | 299.9 | - | - |
| ConceptNet | CONTROLKG train | 5455 | 2898 | 2129 | 16 | 373.0 | - | - |
| | CONTROLKG val. | 606 | 522 | 482 | 16 | 172.3 | | |
| | fruit | 36 | 11 | 37 | 12 | 381.6 | - | - |
| | sun | 36 | 11 | 36 | 12 | 387.5 | - | - |
| | swimming | 40 | 14 | 40 | 15 | 517.8 | - | - |

Table 6: **Examples of KG triplets,** and the best GPT-2 small verbalization for WordNet and ConceptNet.

| Knowledge Graph | | Head | Triplets Relation | Tail | Verbalization |
|---|---|---|---|---|---|
| WordNet | CONTROLKG train | (casserole.n.02, | IsA, | dish.n.01) | "A casserole is a dish" |
| | | (passerby.n.01, | IsA, | pedestrian.n.01) | "A passerby is a type of pedestrian" |
| | | (chorizo.n.01, | IsA, | sausage.n.01) | "A chorizo is a kind of sausage" |
| | CONTROLKG val. | (crate.n.01, | IsA, | box.n.01) | "A crate is a kind of box" |
| | | (magnetometer.n.01, | IsA, | meter.n.02) | "Magnetometer is a type of meter" |
| | | (vaccinee.n.01, | IsA, | patient.n.01) | "A vaccinee is a patient" |
| | communication | (message.n.01, | IsA, | communication.n.02) | "A message is a type of communication" |
| | | (indicator.n.02, | IsA, | signal.n.01) | "An indicator is a type of signal" |
| | | (evidence.n.02, | IsA, | indication.n.01) | "Evidence is an indication" |
| | location | (region.n.01, | IsA, | location.n.01) | "A region is a location" |
| | | (district.n.01, | IsA, | region.n.03) | "A district is a region" |
| | | (expanse.n.03, | IsA, | space.n.02 | "An expanse is a type of space" |
| | representation | (representation.n.02, | IsA, | creation.n.02) | "Representation is a kind of creation" |
| | | (delineation.n.02, | IsA, | drawing.n.02) | "A delineation is a type of drawing" |
| | | (chart.n.02, | IsA, | map.n.01) | "A chart is a map" |
| ConceptNet | CONTROLKG train | (briefcase, | AtLocation, | desk) | "A briefcase is typically placed at a desk" |
| | | (vegetarian, | NotDesires, | meat) | "A vegetarian doesn't crave meat" |
| | | (voting, | Causes, | election) | "A voting can lead to an election" |
| | CONTROLKG val. | (boat, | UsedFor, | sailing) | "A boat is designed for sailing" |
| | | (clothes, ReceivesAction, | | washed) | "Clothes can be washed" |
| | | (jogging, HasPrerequisite, | | energy) | "A jogging requires an energy" |
| | fruit | (fruit, ReceivesAction, | | eaten) | "A fruit can be eaten" |
| | | (wine, | MadeOf, | fruit) | "A wine comprises of a fruit" |
| | | (champagne, | IsA, | wine) | "Champagne is a type of wine" |

## A    DATASET CREATION AND PROCESSING

**TARGETKG**    Given a seed node such as representation in WordNet or fruit in ConceptNet, we sample relations by performing a 3-hop random walk. For example, for the fruit KG shown in Table 6, we start from the seed concept fruit. In the first depth, we retrieve (fruit, ReceivesAction, eaten) and (wine, MadeOf, fruit). In the next depth, we retrieve (champagne, IsA, wine), and so forth for all possible relations. Note that we only sample relations with a single-token tail entity.

Once this connected KG is sampled, we apply two filtering processes. The first one enforces many-to-one relationships in $K_T$ to avoid head entities with multiple tails. The second filtering process reduces the tail-entity imbalance to avoid over-fitting to a small set of tokens. For this, we count the frequency of the tail tokens in the sampled graph and keep at most a quartile amount of triplets with shared tail entities.

Finally, we verbalize TARGETKG graph with the formats that give the lowest perplexity on the pretrained model. We try various relation-specific verbalization templates per knowledge triplet and

pick the one that yields the lowest tail-token perplexity. For example, in the `representation` graph, while the model had lower perplexity with the template "`{h} is a kind of {t}`" for the triplet (`representation.n.02`, `IsA`, `creation.n.02`), it also had lower perplexity with the template "`A {h} is a {t}`" for the triplet (`chart.n.02`, `IsA`, `map.n.01`). Note that this can change for each model size, such as GPT2-small, medium, and large.

In WordNet, a word sense is represented by its lemma, syntactic category, and sense ID (*e.g.*, in `map.n.01`, n for noun and 01 for sense ID). We omit this naming convention from the main paper tables for readability. However, when we do filtering processes we use this sense identification rather than just the lemma.

**CONTROLKG**   To create CONTROLKG, we prioritize not leaking TARGETKG counterfactuals and having a shared CONTROLKG across different TARGETKGs. Therefore, we remove from the complete KG (*e.g.*, for ConceptNet TARGETKGs, the complete LAMA subset of ConceptNet) any triplet that shares the same entities as the union of the TARGETKGs shown in Table 5. For all KG verbalizations, to remove and maintain knowledge that the model is already confident about, we pick the best scoring verbalization for each triplet among several prompt styles.

**CONTROLLM**   We use WikiText-2 (Merity et al., 2017) for the CONTROLLM dataset. We tokenize each entry and then concatenate all of them together. Finally, we group the tokens into chunks of 512. For validation and testing, we use separate held-out sets.

## B   TRAINING AND EVALUATION IMPLEMENTATION

**Hyperparameters**   We use a learning rate of 0.2 with a linear warmup for the first 10% of the training that starts from 1e-10. We optimize with the AdamW optimizer. For equation 6, we set $\lambda_1 = 1.5$ and $\lambda_2 = \lambda_3 = 1$ in all of our our experiments. To encourage the subnetwork to be sparser, we schedule $\lambda_4$ to start at 2 and increase linearly after 50% of the training until it reaches 3.

For GPT2-small, we use a single GPU setting to run the mask training for 40,000 steps. For GPT2-medium and large, we use a three GPU distributed setting and run the mask training for 50,000 steps. All experiments are run on NVIDIA A100 40GB devices.

**Mask Implementation**   As mentioned in §5, during mask learning, we do not mask the embedding, language modeling head, layer-normalization, and bias parameters. We also only learn masks for the top 50% of the transformer layers. We initialize the mask parameters such that, in the first forward pass, each model parameter has a starting masking probability of $\sigma(l_i) = 0.45$, meaning the search is expected to start with an empty knowledge-critical subnetwork (*i.e.*, a subnetwork mask of zeros) and a fully-connected inverse subnetwork (*i.e.*, the full model). Moreover, for the randomly masked baseline, we mask each module (*e.g.*, MLP module at layer 8) at the same sparsity as the corresponding module in the critical subnetwork, which means that the masking is not uniformly done across all layers.

**Dataloaders**   As each TARGETKG is small, at each gradient step, the model sees the complete graph. Therefore, the TARGETKG batch size is the same as the number of triplets (see Table 5). In contrast, CONTROLKG and CONTROLLM datasets have thousands of entries in total. To balance the learning and make it more efficient, we create a dynamic cyclical training dataloader that samples a new batch at each step without replacement. When the dataloader reaches the end of the dataset, it restarts with a new ordering. For the exact batch sizes, please refer to Table 7.

Table 7: **GPU batch size for each dataset and model.**

| Setup | Model | KG | CONTROLLM |
|---|---|---|---|
| Train | GPT-2 Small | 250 | 10 |
|  | GPT-2 Medium | 96 | 4 |
|  | GPT-2 Large | 96 | 4 |
| Eval | GPT-2 Small | 250 | 8 |
|  | GPT-2 Medium | 250 | 8 |
|  | GPT-2 Large | 250 | 8 |

Table 9: **Success metrics for different percentages of upper layers masked in GPT-2 small,** averaged over four KGs and two seeds with [min, max] values denoted in brackets. The arrows ($\uparrow$,$\downarrow$) show the desired value for the metric.

| Masked Layer Choice | Percentage Masked | Sparsity ($\uparrow$) | TARGETKG $\Delta$ PPL($\uparrow$) | CONTROLKG $\Delta$ PPL ($\downarrow$) | CONTROLLM $\Delta$ PPL ($\downarrow$) | # of checkpoints |
|---|---|---|---|---|---|---|
| 0-11 | 100% | 95.6 [94.9, 96.6] | 242.7 [-26.6, 1254.3] | 11.8 [6.7, 15.9] | 1.3 [1.0, 1.7] | 1.1 |
| 3-11 | 75% | 97.3 [94.4, 98.3] | 669.7 [-8.7, 2119.7] | -1.4 [-8.3, 10.8] | 1.0 [0.5, 2.9] | 76.9 |
| 6-11 | 50% | 98.6 [97.1, 99.2] | 870.4 [38.7, 2665.1] | 0.4 [-2.6, 4.0] | 0.5 [0.3, 1.0] | 104.1 |
| 9-11 | 25% | 99.2 [98.0, 99.7] | 1185.0 [62.3, 4787.0] | 4.2 [0.1, 9.5] | 0.4 [0.0, 0.8] | 103.2 |

**Best Checkpoint Selection** We iteratively select the best checkpoint, starting with strict criteria on the maintenance datasets and gradually loosening them. We check whether any checkpoints satisfy the first set of criteria limits shown in Table 8. The checkpoints need to have a TARGETKG $\Delta$ PPL above the mentioned floor

Table 8: **Selection limit for each success criteria.**

| Iteration | TARGETKG $\Delta$ PPL Floor | CONTROLKG $\Delta$ PPL Ceiling | CONTROLLM $\Delta$ PPL Ceiling |
|---|---|---|---|
| 1 | 35.0 | 5.0 | 1.0 |
| 2 | 40.0 | 7.0 | 2.0 |
| 3 | 40.0 | 10.0 | 3.0 |
| 4 | 50.0 | 15.0 | 4.0 |

and maintenance $\Delta$ PPL below the mentioned ceiling. If the set of checkpoints retrieved is empty, we select from the next set of limits. If none of the iterations are successful, we pick the last checkpoint as the best one.

## C  MASKED LAYER CHOICE STUDY

Layer-wise model probing analyses have shown that the first layers of transformer language models encode representations crucial for low-level linguistic tasks and features that may be a prerequisite for knowledge modeling (Tenney et al., 2019; Liu et al., 2019). Researchers have also shown that knowledge is not only contained in the final few layers (Wallat et al., 2020). Therefore, for our datasets, we investigate how masking different percentages of upper dense layers can affect the success criteria defined for a knowledge-critical subnetwork. We look at masking the top 25%, 50%, 75%, and 100% of the model.

In Table 9, we observe that masking all dense layers in transformer blocks (100%) can affect the maintenance criteria significantly. CONTROLKG perplexity difference is smaller when masking fewer layers, confirming that lower layers may have imperative representation to knowledge modeling. As the values for the different criteria are similar for masking the top 25% and 50%, we choose to use the top 50% masking approach to increase the masking coverage for all of our experiments.

## D  ADDITIONAL SUBNETWORK DISCOVERY RESULTS

In this section, we provide additional metrics for subnetwork discovery results and non-aggregated results for the randomly masked baseline.

**Rank and Log Probabilities** In addition to the perplexity difference between the remaining model and the complete pretrained model (*i.e.*, $\Delta$ PPL), we provide rank and log probability differences ($\Delta$ Rank and $\Delta$ LogProb), where rank is the rank of the gold tail token in the prediction distribution. We observe in Table 10 the same trend as the perplexity differences. On average, removing the subnetwork increases the rank of the gold tail token and decreases the log probability. In contrast, the randomly masked baseline does not increase the TARGETKG rank significantly and does not maintain CONTROLKG rank to the same extent as the critical subnetwork.

**Model Scale** We include the individual KG results for larger models in Table 11. While individual results on GPT2-medium are not as sparse and effective as the small and large variants, it is still more significant than randomly masking the model at the same sparsity.

**Randomly Masked Baseline** We provide the non-aggregated randomly masked baseline results for GPT2-small in Table 12 and for larger models in Table 13. We notice that KGs where the pretrained

Table 10: **Subnetwork discovery rank and log probability results for GPT-2 small,** averaged over three seeds. $\Delta$ Metric = Metric($f(x, \tilde{m} \odot \theta)$) - Metric($f(x, \theta)$) for Rank and LogProb. Random is an average of randomly masked baselines at the same sparsity levels as the discovered knowledge-critical subnetworks for each KG-seed pair. Note that non-zero values may be rounded to 0.0 as we round to one decimal place.

| | Knowledge Graph | TARGETKG $\Delta$ Rank ($\uparrow$) | CONTROLKG $\Delta$ Rank ($\downarrow$) | TARGETKG $\Delta$ LogProb ($\downarrow$) | CONTROLKG $\Delta$ LogProb ($\uparrow$) |
|---|---|---|---|---|---|
| WordNet | building.n.01 | 83.7 [12.8, 168.3] | 1.1 [-1.1, 2.9] | -0.7 [-1.2, -0.2] | 0.0 [0.0, 0.1] |
| | communication.n.02 | 117.0 [94.5, 134.9] | 0.6 [0.1, 1.0] | -0.7 [-1.0, -0.5] | 0.0 [0.0, 0.0] |
| | change.n.01 | 139.1 [0.4, 409.8] | 0.4 [0.1, 0.6] | -1.4 [-2.6, -0.3] | 0.0 [0.0, 0.0] |
| | statement.n.01 | 154.5 [1.6, 353.8] | 0.8 [-0.6, 2.8] | -0.6 [-0.9, -0.3] | 0.0 [0.0, 0.0] |
| | location.n.01 | 344.9 [188.4, 527.6] | 3.6 [2.8, 5.0] | -1.6 [-2.0, -1.2] | 0.0 [-0.1, 0.0] |
| | representation.n.02 | 38.1 [12.8, 57.8] | 3.4 [2.7, 4.4] | -0.7 [-1.0, -0.4] | 0.0 [-0.1, 0.0] |
| | magnitude.n.01 | 1368.7 [978.0, 1698.2] | 0.0 [-0.3, 0.1] | -2.1 [-2.3, -1.9] | 0.0 [0.0, 0.0] |
| | Random | 12.0 [-0.1, 25.5] | 2.7 [-0.1, 8.1] | -0.1 [-0.3, 0.0] | -0.2 [-0.5, 0.0] |
| | Average | 320.9 [38.1, 1368.7] | 1.4 [0.0, 3.6] | -1.1 [-2.1, -0.6] | 0.0 [0.0, 0.0] |
| ConceptNet | fruit | 1164.9 [98.9, 2880.1] | 1.8 [0.1, 3.5] | -1.0 [-1.6, -0.6] | 0.0 [0.0, 0.0] |
| | sun | 331.7 [225.9, 415.8] | 1.6 [1.4, 1.7] | -1.2 [-1.4, -0.9] | 0.0 [0.0, 0.0] |
| | swimming | 411.6 [34.5, 685.6] | 1.4 [0.8, 1.9] | -0.4 [-0.5, -0.4] | 0.0 [0.0, 0.0] |
| | Random | 11.4 [2.1, 20.0] | 5.5 [4.2, 7.3] | 0.0 [-0.1, 0.0] | -0.1 [-0.1, -0.1] |
| | Average | 636.1 [331.7, 1164.9] | 1.6 [1.4, 1.8] | -0.9 [-1.2, -0.4] | 0.0 [0.0, 0.0] |

Table 11: **Subnetwork discovery results on larger models per KG,** averaged over two seeds. Random is an average of randomly masked baselines at the same sparsity levels as the discovered knowledge-critical subnetworks for each KG-seed pair.

| Model Size | Knowledge Graph | Sparsity ($\uparrow$) | TARGETKG $\Delta$ PPL ($\uparrow$) | CONTROLKG $\Delta$ PPL ($\downarrow$) | CONTROLLM $\Delta$ PPL ($\downarrow$) |
|---|---|---|---|---|---|
| Medium | communication.n.02 | 99.5 [99.4, 99.7] | 139.9 [-2.5, 282.3] | -0.1 [-1.5, 1.3] | 0.1 [0.0, 0.1] |
| | location.n.01 | 95.0 [94.2, 95.8] | 432.2 [48.0, 816.3] | 3.7 [3.5, 3.8] | 0.8 [0.8, 0.9] |
| | representation.n.02 | 94.8 [91.9, 97.7] | 194.6 [139.4, 249.8] | 4.0 [3.8, 4.2] | 1.2 [0.5, 1.8] |
| | Random | 96.4 [94.8, 99.5] | 32.1 [5.0, 55.6] | 9.2 [1.8, 15.9] | 3.0 [0.3, 4.9] |
| | Average | 96.4 [94.8, 99.5] | 255.6 [139.9, 432.2] | 2.5 [-0.1, 4.0] | 0.7 [0.1 , 1.2] |
| Large | communication.n.02 | 95.9 [95.4, 96.4] | 2013.1 [144.5, 3881.8] | 6.8 [3.6, 10.0] | 0.6 [0.5, 0.6] |
| | location.n.01 | 99.6 [99.4, 99.7] | 1963.1 [1543.1, 2383.1] | 1.9 [0.6, 3.3] | 0.0 [-0.0, 0.0] |
| | representation.n.02 | 99.2 [99.1, 99.4] | 13363.6 [3042.2, 23685.0] | 0.9 [0.2, 1.7] | 0.0 [0.0, 0.1] |
| | Random | 98.2 [95.9, 99.6] | 6.8 [4.8, 7.8] | 2.9 [0.7, 7.3] | 0.8 [0.2, 2.1] |
| | Average | 98.2 [95.9, 99.6] | 5779.9 [1963.1, 13363.6] | 3.2 [0.9, 6.8] | 0.2 [0.0, 0.6] |

model perplexity is already low (see Table 5) seem not to be as affected by a random subnetwork removal as those that have a higher initial perplexity.

# E ADDITIONAL PARAMETER REMOVAL AND ADDITION ANALYSIS

We include additional KG results for removing and adding parameters to the remaining model (Figure 3), specifically for location and communication. As mentioned in §6.3, the TARGETKG recovery extent seems to depend on the degree the knowledge was suppressed in the first place. If the model forgot to express the knowledge by a significant amount, removing further parameters decreases TARGETKG $\Delta$ PPL slowly and not as significantly when the critical subnetwork is compared to the randomly masked baseline (*e.g.*, representation and location). On the other hand, the communication-critical subnetwork was at sparsity $\approx$99% with a TARGETKG perplexity difference of 166, and further removing parameters decreases the perplexity difference on average to 60 at 92% sparsity, indicating that its suppression may have overfit to a particular configuration. However, we note that even when $\Delta$ PPL drops to 60, this difference is still higher than a randomly masked baseline average.

Table 12: **Success metrics on the randomly masked baseline,** averaged over three seeds.

| Knowledge Graph | | Sparsity (↑) | TARGETKG Δ PPL(↑) | CONTROLKG Δ PPL (↓) | CONTROLLM Δ PPL (↓) |
|---|---|---|---|---|---|
| WordNet | building | 98.4 [97.4, 99.3] | 5.8 [3.0, 10.2] | 14.8 [3.0, 26.2] | 2.8 [1.0, 5.2] |
| | communication | 99.2 [99.0, 99.3] | 5.0 [-2.4, 10.1] | 4.6 [0.3, 7.6] | 1.2 [1.0, 1.4] |
| | change | 98.4 [98.0, 99.1] | 33.9 [25.7, 43.3] | 24.2 [16.2, 36.2] | 2.0 [1.3, 2.6] |
| | statement | 98.2 [96.3, 99.2] | 15.6 [-0.3, 34.6] | 0.0 [-3.5, 3.4] | 3.3 [1.2, 6.8] |
| | location | 99.0 [98.8, 99.1] | 18.8 [-15.8, 55.8] | 0.2 [-7.5, 4.6] | 1.6 [1.3, 1.9] |
| | representation | 98.1 [97.1, 98.8] | 48.8 [11.4, 80.3] | 46.2 [30.6, 66.2] | 3.0 [1.6, 4.5] |
| | magnitude | 99.0 [98.6, 99.3] | 41.9 [21.1, 70.4] | 12.3 [-0.2, 29.2] | 1.6 [0.8, 2.0] |
| | Average | 98.6 [98.1, 99.2] | 24.3 [5.0, 48.8] | 14.6 [0.0, 46.2] | 2.2 [1.2, 3.3] |
| ConceptNet | fruit | 99.2 [99.1, 99.4] | 13.7 [-12.6, 35.7] | 12.4 [-0.6, 19.7] | 1.3 [0.9, 1.7] |
| | sun | 99.2 [99.0, 99.3] | 29.4 [11.0, 39.0] | 17.2 [0.3, 36.3] | 1.5 [1.1, 1.8] |
| | swimming | 99.0 [98.8, 99.2] | 19.8 [-22.1, 42.6] | 14.1 [2.3, 30.2] | 1.7 [1.2, 2.4] |
| | Average | 99.1 [99.0, 99.2] | 21.0 [13.7, 29.4] | 14.6 [12.4, 17.2] | 1.5 [1.3, 1.7] |

Table 13: **Success metrics on larger randomly masked models per KG,** averaged over two seeds.

| Model Size | Knowledge Graph | Sparsity (↑) | TARGETKG Δ PPL (↑) | CONTROLKG Δ PPL (↓) | CONTROLLM Δ PPL (↓) |
|---|---|---|---|---|---|
| Medium | communication.n.02 | 99.5 [99.4, 99.7] | 5.0 [1.9, 8.1] | 1.8 [1.5, 2.1] | 0.3 [0.3, 0.4] |
| | location.n.01 | 95.0 [94.2, 95.8] | 55.6 [35.2, 76.1] | 15.9 [11.6, 20.2] | 3.8 [3.3, 4.4] |
| | representation.n.02 | 94.8 [91.9, 97.7] | 35.8 [18.0, 53.7] | 9.9 [9.6, 10.3] | 4.9 [1.5, 8.3] |
| | Average | 96.4 [94.8, 99.5] | 32.1 [5.0, 55.6] | 9.2 [1.8, 15.9] | 3.0 [0.3, 4.9] |
| Large | communication.n.02 | 95.9 [95.4, 96.4] | 7.8 [7.2, 8.4] | 7.3 [6.6, 8.0] | 2.1 [1.8, 2.5] |
| | location.n.01 | 99.6 [99.4, 99.7] | 4.8 [3.6, 6.0] | 0.7 [0.5, 0.9] | 0.2 [0.1, 0.3] |
| | representation.n.02 | 99.2 [99.1, 99.4] | 7.8 [5.4, 10.1] | 0.8 [-0.2, 1.8] | 0.2 [0.2, 0.3] |
| | Average | 98.2 [95.9, 99.6] | 6.8 [4.8, 7.8] | 2.9 [0.7, 7.3] | 0.8 [0.2, 2.1] |

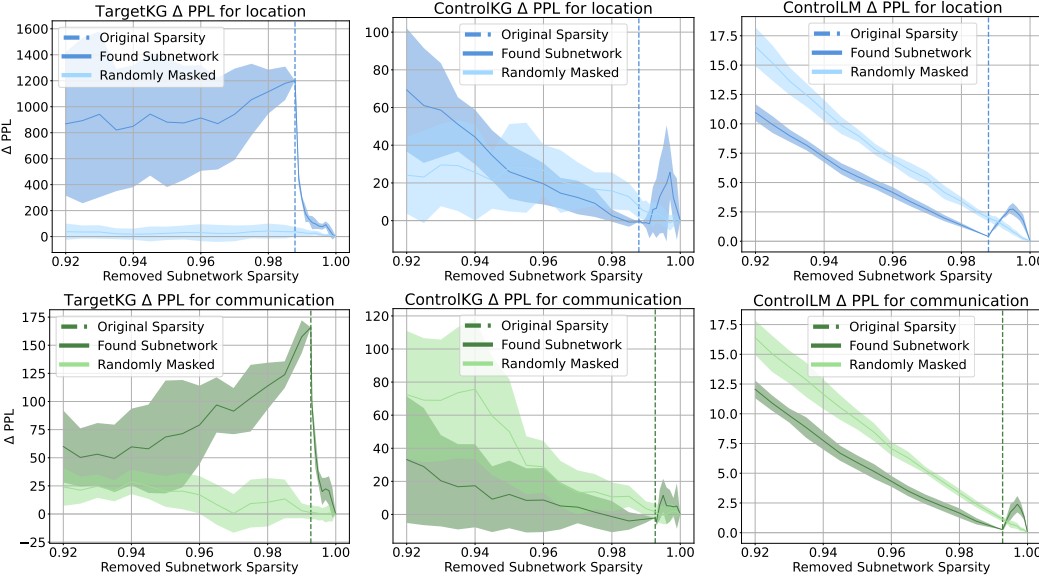

Figure 3: **Removing and adding parameters to the remaining model,** averaged over five seeds, with standard deviation depicted as the filled area around the average curves. The $x$-axis is the removed subnetwork sparsity. The $y$-axis is the Δ PPL = PPL($f(x, \tilde{\boldsymbol{m}} \odot \boldsymbol{\theta})$) - PPL($f(x, \boldsymbol{\theta})$) for the different datasets. Vertical dashed lines show the original sparsity of the critical subnetwork. The darker curve is the outcome starting from the critical subnetwork, whereas the lighter curve is from a randomly masked model at the same sparsity.

Table 14: **Accuracy on downstream CommonsenseQA task,** averaged over three seeds. Ours refers to removing the critical subnetwork. Random refers to removing a random subnetwork at the same sparsity as the critical subnetwork.

| Method | Subnetwork | Dev | Test | Filtered | Half 1 | Half 2 |
|---|---|---|---|---|---|---|
| **Head Tuning** | Full | 38.63 | 38.33 | 37.19 | 37.94 | 36.44 |
| | Random (Half 1) | -0.87 | -4.83 | -4.89 | -1.75 | -8.00 |
| | Ours (Half 1) | -1.45 | -4.83 | -11.02 | -15.11 | -6.95 |
| | Random (Half 2) | -0.46 | -4.16 | -4.27 | -2.30 | -6.22 |
| | Ours (Half 2) | -1.99 | -6.80 | -8.61 | -7.28 | -9.93 |
| **LoRA** | Full | 50.04 | 48.64 | 48.67 | 48.44 | 48.90 |
| | Random (Half 1) | -1.39 | -0.08 | -1.84 | -0.93 | -2.75 |
| | Ours (Half 1) | -0.54 | -2.33 | -2.48 | -2.95 | -2.02 |
| | Random (Half 2) | -1.23 | -0.96 | -1.75 | -0.93 | -2.57 |
| | Ours (Half 2) | -0.05 | -1.93 | -3.77 | -3.14 | -4.39 |
| **Full Finetuning** | Full | 44.61 | 42.33 | 42.79 | 44.01 | 41.58 |
| | Random (Half 1) | +0.08 | -0.62 | -0.36 | -2.94 | +2.19 |
| | Ours (Half 1) | +0.50 | -1.34 | -0.46 | -5.33 | +4.39 |
| | Random (Half 2) | -1.01 | 0.00 | +0.74 | -1.65 | +3.11 |
| | Ours (Half 2) | -0.11 | -0.75 | -0.27 | -0.92 | +0.36 |

## F  ADDITIONAL DETAILS ON DOWNSTREAM TASK TRANSFER

To learn a mask for a set of ConceptNet relations, we need to verbalize them with a relation-specific prompt. As described in §6.4, CommonsenseQA questions are not explicitly annotated with a relation. However, they were constructed with ConceptNet such that each question's head concept relates to four of the tail answers with the same relation. This does not apply to the fifth answer, as crowd workers created them. Therefore, to retrieve the relations, we iterate through the questions and check if any relations with the question head concept and correct tail answer exist in the LAMA and Commonsense Knowledge Base Completion subsets of ConceptNet (Li et al., 2016; Petroni et al., 2019). If it does and has only one relation, we choose that relation. If it has multiple relations, we take the union of relations between the head concept and the distractor tail answers and intersect that with the correct tail triplets. If the intersection is a set larger than one element, we we choose one relation at random. Out of the 1221 test questions, only 572 have a single-token correct answer, and we could only find the corresponding relation to 363 questions, which is our filtered test set.

For the MCQA head, we use the Huggingface Double Heads model.[6] In addition to the language modeling head, this model adds a parallel multiple-choice classification head. The MCQA head takes as input the last sequence output. To finetune the MCQA model, we use three kinds of fine-tuning. The first one is **Head Tuning**, in which the model parameters are frozen, but the MCQA head is not. The second method is **LoRA** (Hu et al., 2022), which is a parameter-efficient finetuning method. Similar to the head tuning method, LoRA freezes the model parameters and instead inserts trainable rank decomposition parameters in each transformer layer. We use a rank of 16 for all LoRA experiments. Finally, we also try **Full Finetuning**, in which all model parameters are tuned. To remove a subnetwork, we manually set the knowledge-critical parameters to 0. Therefore, the value of these parameters can change during full finetuning.

In addition, we also verify whether learning a mask for one randomly selected half of the filtered test set (Half 1) corrupts downstream task transfer for a distinct half (Half 2), where there are no triplet overlaps. We find in Table 14 that, on average, the accuracy on the triplets the mask was trained for is less by 3.6% than the held-out half.

---

[6]https://huggingface.co/docs/transformers

Table 15: **Expression loss study,** averaged across three KGs and two seeds. Random is an average of randomly masked baselines at the same sparsity levels as the discovered knowledge-critical subnetworks for each KG-seed pair.

| Objective Combination | Sparsity ($\uparrow$) | TARGETKG $\Delta$ PPL ($\uparrow$) | CONTROLKG $\Delta$ PPL ($\downarrow$) | CONTROLLM $\Delta$ PPL ($\downarrow$) |
|---|---|---|---|---|
| Expression-only | 99.8 [99.7, 99.9] | 154.0 [105.4, 181.2] | 83.5 [-4.2, 234.9] | 4.0 [2.6, 6.2] |
| Our Method + Expression | 95.7 [93.8, 96.7] | 909.2 [107.2, 2421.7] | -0.5 [-4.7, 5.1] | 1.0 [0.8, 1.4] |
| Our Method | 98.6 [97.8, 99.1] | 378.1 [74.3, 834.9] | 1.6 [-0.7, 4.0] | 0.5 [0.3, 0.8] |

## G  ALTERNATIVE OBJECTIVE: IS EXPRESSING KNOWLEDGE ENOUGH TO BE A KNOWLEDGE-CRITICAL SUBNETWORK?

We defined *knowledge-critical subnetworks* as being *responsible* for a model's ability to express certain pieces of knowledge, validated by an increase in perplexity when that subnetwork is removed from the model. However, another way to extract a knowledge-critical subnetwork might be to learn a mask over the network that minimizes the negative loglikelihood of all $x \in$ TARGETKG:

$$\mathcal{L}_{\text{express}} = -\sum_x \log(f(x, \boldsymbol{m} \odot \boldsymbol{\theta})) \tag{7}$$

In Table 15, we compare subnetworks extracted in this manner (*i.e.*, Expression-only) with those of our main method, as well as those of a combination of these objectives: $\mathcal{L}_{\text{final}} + \lambda_5 \mathcal{L}_{\text{express}}$. Interestingly, we find that the Expression-only setting can learn a mask for a *highly* sparse subnetwork, which, when removed from the full model, also significantly increases perplexity on TARGETKG. However, this subnetwork also struggles to maintain perplexity on CONTROLKG, indicating it may encode abilities crucial for expressing *any* set of relational knowledge. Adding the expression loss to our joint objective mitigates this issue, but reduces subnetwork sparsity by a significant margin ($\sim$4%), indicating that the Expression-only loss may discover spurious subnetworks that are not actually *knowledge-critical* — they are not *responsible* for the expression of the knowledge when they are entangled in the full model, though their parameters may compute a function that expresses it.

## H  PARAPHRASE GENERALIZATION

To make sure that the remaining model is not overfit to the specific prompt template we used during training and that TARGETKG knowledge is removed with other prompting styles, we evaluate the remaining models on paraphrases of the knowledge triplet. We run an experiment where we evaluate $\Delta$ PPL of TARGETKG and CONTROLKG for all prompt styles, excluding the one used in the training process, which had the lowest tail-token perplexity on the pretrained model. This results in 20 other distinct templates that differ either by the relation token, by the entity articles, or with a space put in front of the sentence.

Table 16: **Success metrics on paraphrases.** The arrows ($\uparrow$,$\downarrow$) show the desired value for the metric. $\Delta$ PPL = PPL($f(x, \tilde{\boldsymbol{m}} \odot \boldsymbol{\theta})$) - PPL($f(x, \boldsymbol{\theta})$).

| Knowledge Graph | TARGETKG $\Delta$ PPL ($\uparrow$) | CONTROLKG $\Delta$ PPL ($\downarrow$) |
|---|---|---|
| communication | 492.4 | 3.0 |
| location | 684.9 | -33.0 |
| representation | 916.4 | -6.9 |

Note that the average perplexity of the original model on the *worse* paraphrases is 3231 for TARGETKG and 2368 for CONTROLKG averaged across three KGs (`representation`, `location`, and `communication`).

The success metric results do not change when using other prompt styles, as seen in Table 16 compared to Table 1. We note that the model perplexity on CONTROLKG paraphrases is a bit lower than the format used during training, likely because the default perplexity is higher on other templates. Because it is higher than the original templates for CONTROLKG, the model may generalize its maintenance of CONTROLKG to a greater degree on these suboptimal templates. In summary, our method generalizes to also suppress alternate prompts of TARGETKG.

## I  SEED-BASED ANALYSIS

We investigate the stability of subnetwork discovery under seed-based variance. We also explore whether composing subnetworks from different seeds could increase the suppression effect while still fulfilling the rest of the success criteria.

**Seed-based Variance**   Prior work shows that subnetworks identified under distinct random seeds may differ with a large variance (Csordás et al., 2021). ~~Here, we analyze how much this difference can be in localizing knowledge-specific subnetworks.~~ We inspect how subnetworks from the best checkpoints for three random seeds overlap for an individual TARGETKG. We use Jaccard similarity, or intersection over union (IoU), as the overlap metric. In Figure 4, we plot a Venn diagram of parameter overlap for each knowledge graph. We can see that, on average, when using IoU, only around 3.7% of the unioned subnetwork parameters overlap across the three seeds (3.76% for `location`, 3.8% for `communication`, and 3.5% for `representation`), meaning the subnetworks identified under different random seeds vary, which complies with prior works' analysis. Across layers, the IoU is also similarly low with a ~~slightly~~ higher overlap for the final attention layer masks ($\approx$10%) as shown in Figure 5.

**Subnetwork Composition**   We combine masks of three seeds in their intersection, their floral intersection (intersection unioned with each intersection of two seeds), and overall union to measure the effect on $\Delta$ PPL for TARGETKG, CONTROLKG, and CONTROLLM. We average the results over three KGs (`representation`, `location`, and `communication`).

Table 17: **Composing subnetworks across seeds,** averaged across three KGs. Original stands for the individual subnetwork removal average across the same three seeds and KGs.

| Mask Pattern | Sparsity ($\uparrow$) | TARGETKG $\Delta$ PPL ($\uparrow$) | CONTROLKG $\Delta$ PPL ($\downarrow$) | CONTROLLM $\Delta$ PPL ($\downarrow$) |
|---|---|---|---|---|
| Original | 98.8 | 379.1 | 0.7 | 0.4 |
| Union | 96.9 | 4013.3 | 31.0 | 5.3 |
| Floral | 99.5 | 84.8 | 8.9 | 1.4 |
| Intersect | 99.9 | 10.6 | 6.6 | 2.4 |

In Table 17, we observe that removing the intersection and floral intersection of the subnetworks does not increase TARGETKG $\Delta$PPL. On the other hand, removing the union of the subnetworks increases the TARGETKG perplexity difference significantly larger than the original results. However, combining the subnetworks and removing them increases $\Delta$PPL on maintenance datasets more than using an individual seed's subnetwork, as seen in the original results. We note that the increase in the $\Delta$PPL on maintenance datasets matches the increase we get when removing an equally sparse random subnetwork (see Table 1). Therefore, it may be possible to naively combine subnetworks, however, they may not guarantee the maintenance criteria to the same extent. A future idea could be to continue optimizing for the subnetwork mask by initializing it as the union of the subnetworks to see if more robust suppression can be achieved.

## J  KNOWLEDGE-BASED ANALYSIS

This section examines the overlap of subnetworks across different KGs for the same seed. This contrasts with the previous section that studies the overlap of subnetworks across different seeds for the same KG. Similarly, we use Jaccard similarity, or intersection over union (IoU), as the overlap metric. We also explore whether composing subnetworks for different KGs from the same seed could suppress all of the TARGETKGs.

**Knowledge-based Variance**   In Figure 6, we plot a Venn diagram of parameter overlap for each seed across different TARGETKGs. We can see that, on average, when using IoU, only around 3.56% of the unioned subnetwork parameters overlap across the three seeds (4.08% for `seed 735`, 4.01% for `seed 1318`, and 2.65% for `seed 84`), meaning the subnetworks identified under different KGs. Across layers, the IoU is also similarly low with a significantly higher overlap for the final attention layer masks ($\approx$12%) as shown in Figure 7.

**Subnetwork Composition**   We combine masks of three KGs for the same seed in their intersection, their floral intersection (intersection unioned with each intersection of two KGs), and overall union to

measure the effect on $\Delta$ PPL for TARGETKG, CONTROLKG, and CONTROLLM. We average the results over three seeds (735, 1318, and 84).

Similar to the findings in composing sub-networks for different seeds, Table 18 shows that composing subnetworks for different KGs increases the $\Delta$PPL on TAR-GETKG when using their union. However, removing the union of the subnetworks also has higher perplexity differences on main-tenance datasets than using an individual KG's subnetwork, as seen in the original results. Once again, this $\Delta$PPL increase on the maintenance datasets matches the dif-

Table 18: **Composing subnetworks across KGs,** aver-aged across three seeds. Original stands for the individ-ual subnetwork removal average across the same three seeds and KGs.

| Mask Pattern | Sparsity ($\uparrow$) | TARGETKG $\Delta$ PPL ($\uparrow$) | CONTROLKG $\Delta$ PPL ($\downarrow$) | CONTROLLM $\Delta$ PPL ($\downarrow$) |
|---|---|---|---|---|
| Original | 98.8 | 379.1 | 0.7 | 0.4 |
| Union | 96.9 | 1984.4 | 44.9 | 4.7 |
| Floral | 99.5 | 4.9 | 4.5 | 1.1 |
| Intersection | 99.9 | 7.9 | 3.7 | 2.1 |

ference we would observe using an equally sparse random subnetwork. Therefore, while subnetworks of different KGs may be composable to fortify the suppression effect, they may not guarantee the maintenance criteria to the same extent as the individual subnetworks.

## K   STRUCTURAL ANALYSIS

In this section, we investigate the structure of the removed knowledge-critical subnetworks by looking at their density across different layer types (Figure 8), and more specifically, across different attention heads (Figure 9) and the $W_q$, $W_k$, and $W_v$ matrices in attention sublayers (Figure 10.

Layer depth-wise, we observe that the subnetwork is consistently most dense around the first and final masked transformer blocks, which are layers 7 and 12 in Figure 8. Specifically, layer type-wise, we find that knowledge-critical subnetworks are most dense in the attention sublayers for layer 7 and layer 12 (`Attn-Out` and `Attn-`$W_q, W_k, W_v$). However, in the middle layers, the most dense layer types are Feed-Forward networks, particularly their first linear sublayer (`FF-1`).

In addition, we have not found any complete columns or rows that were dense in the critical subnetworks. This means no input or output neuron features get completely removed when the critical subnetwork is removed. Therefore, the masked region may not be working to zero-out the knowledge by turning specific features off, which would counter the prevailing view that neuron-level changes are necessary for mechanistic interventions (Dai et al., 2022; Meng et al., 2022).

When we investigated attention heads and $W_q$, $W_k$, and $W_v$ masks in detail for 3 KGs and 3 seeds, we found that head 10 in layer 7, and heads 1 and 9 in layer 12 are significantly dense. Moreover, the V mask is consistently the most dense across the three attention $W_q$, $W_k$, and $W_v$ masks. Therefore, while the subnetworks do not have a significant IoU, as demonstrated by the seed-based (Appendix I) and the KG-based analyses (Appendix J), the subnetworks still tend to be dense in similar layer types at similar layer depths.

Parameter overlap for different seeds and same KG "communication.n.02"

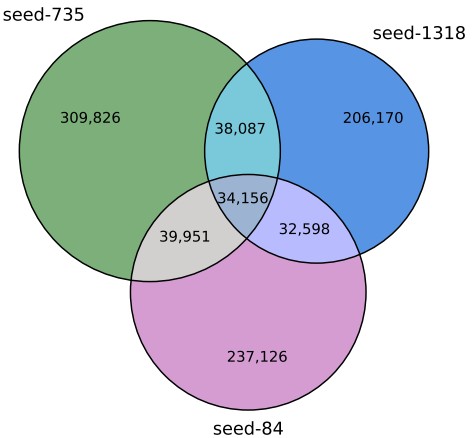

Parameter overlap for different seeds and same KG "location.n.01"

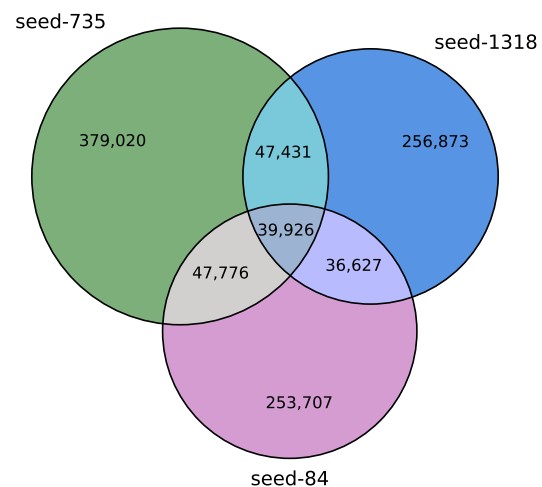

Parameter overlap for different seeds and same KG "representation.n.02"

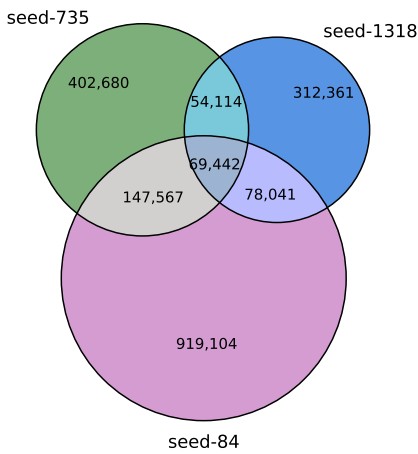

Figure 4: Venn diagrams for parameter overlap of three subnetworks identified under three different random seeds, for each KG representation, location, and communication.

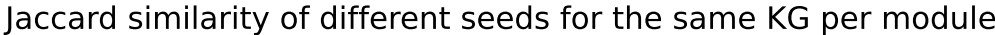

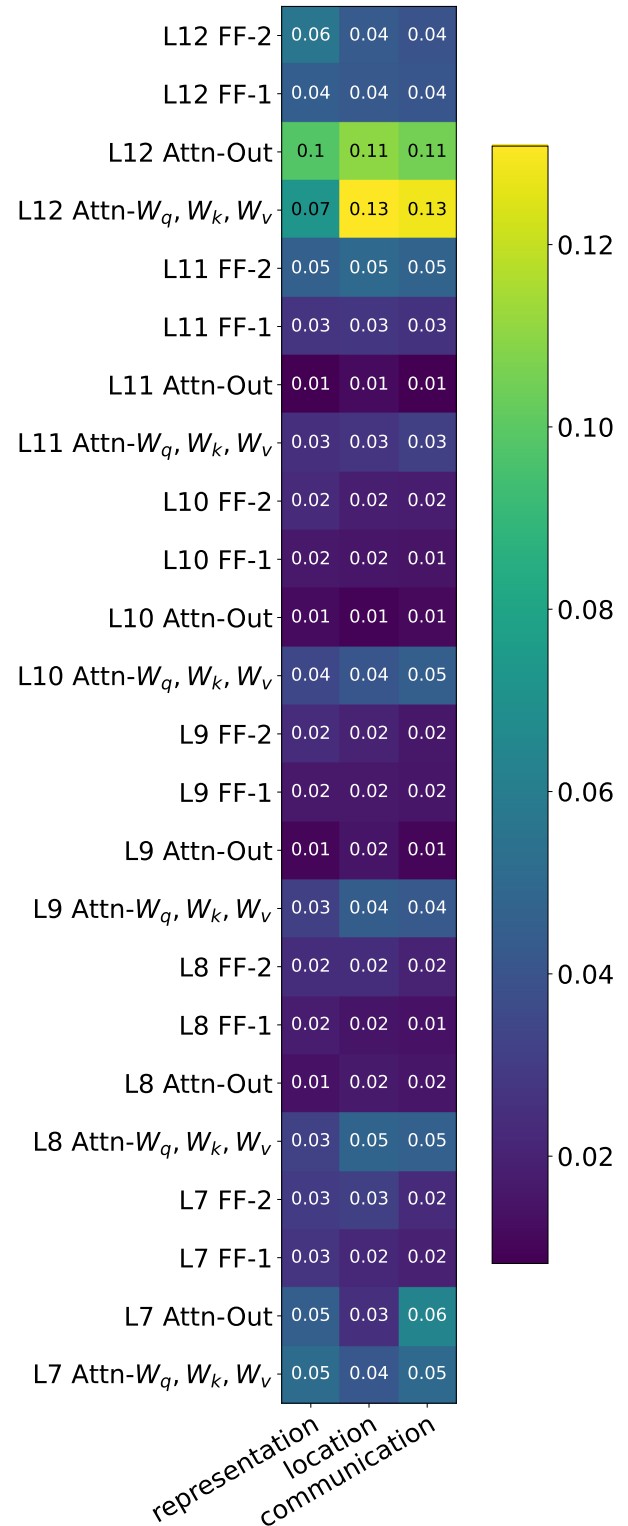

Figure 5: **Jaccard similarity of different seed masks for the same KG,** (representation, location, and communication). The brighter the color, the higher the Intersection over Union.

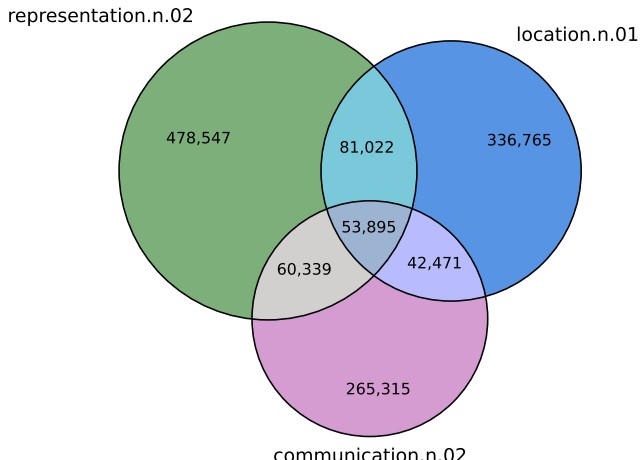

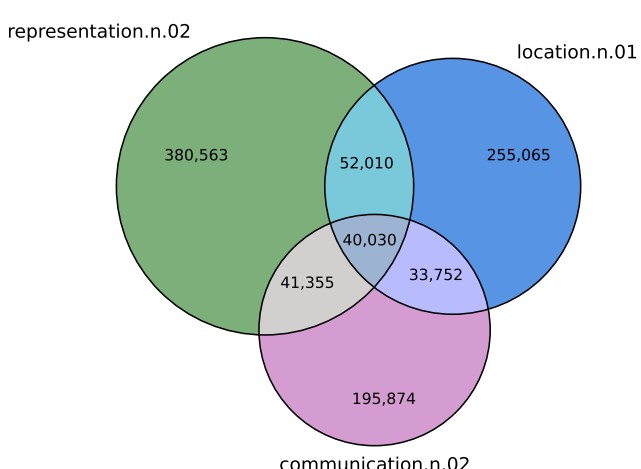

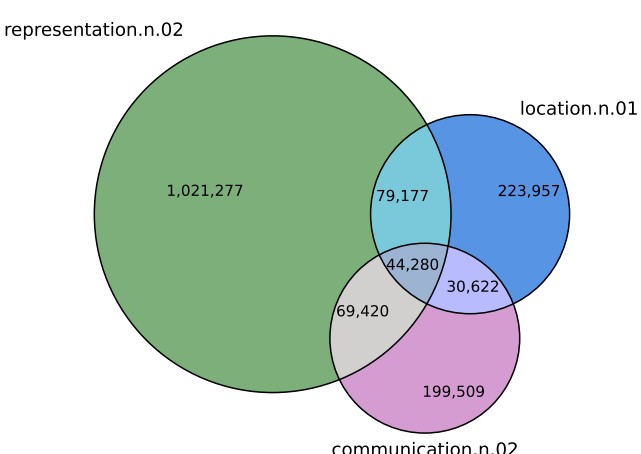

Figure 6: Venn diagrams for parameter overlap of three subnetworks identified under three different KGs, for each seed 735, 1318, and 84.

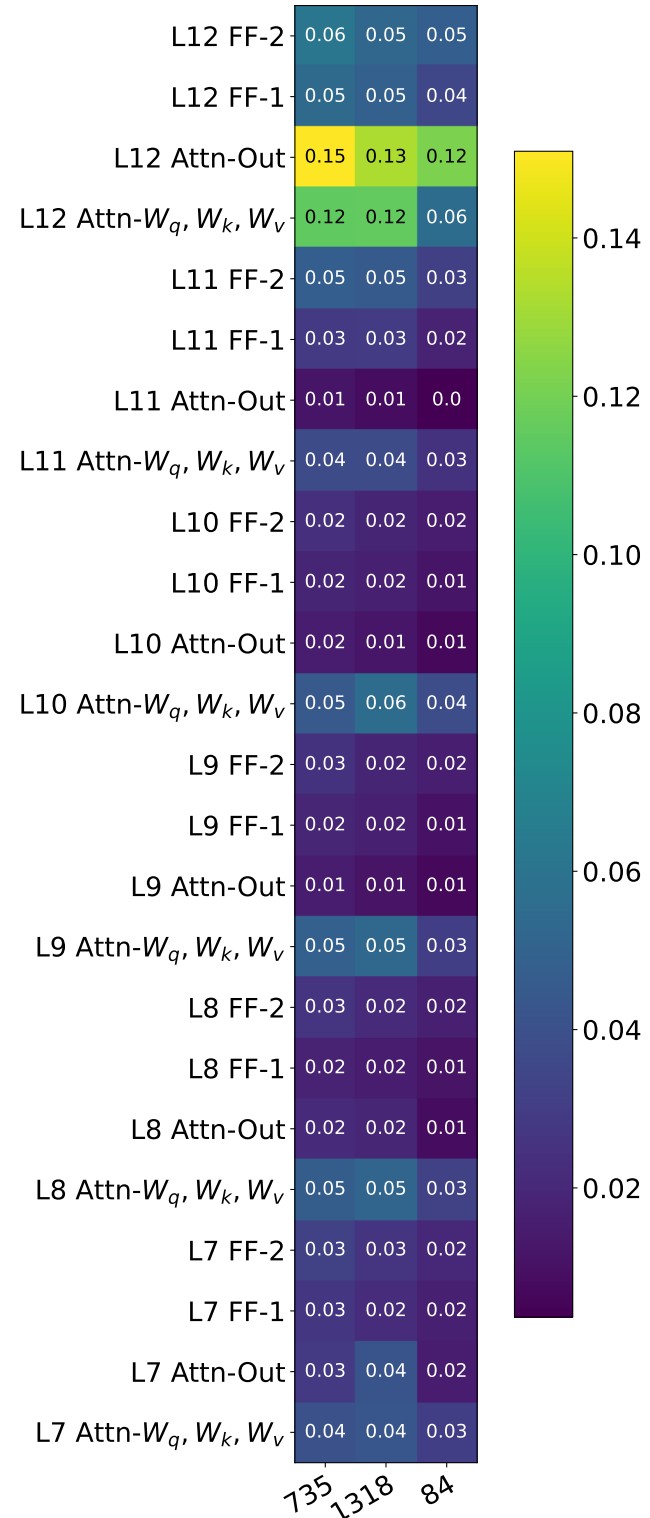

Figure 7: **Jaccard similarity of different KG masks for the same seed,** (735, 1318, and 84). The brighter the color, the higher the Intersection over Union.

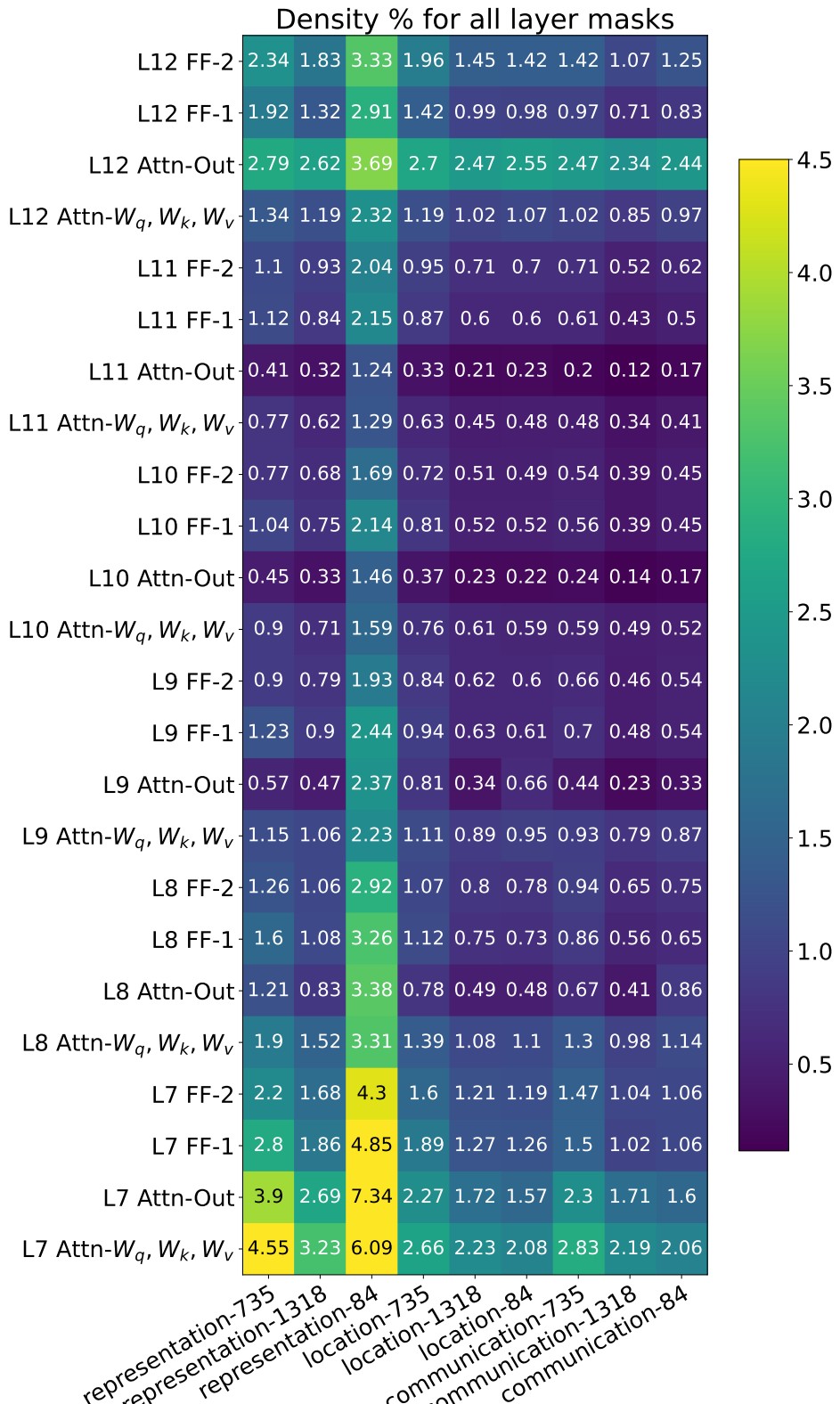

Figure 8: **Average module mask density**, for different KGs ( representation, location, and communication) and seeds. Reported in percentage (%). The brighter the color, the higher the removed mask density.

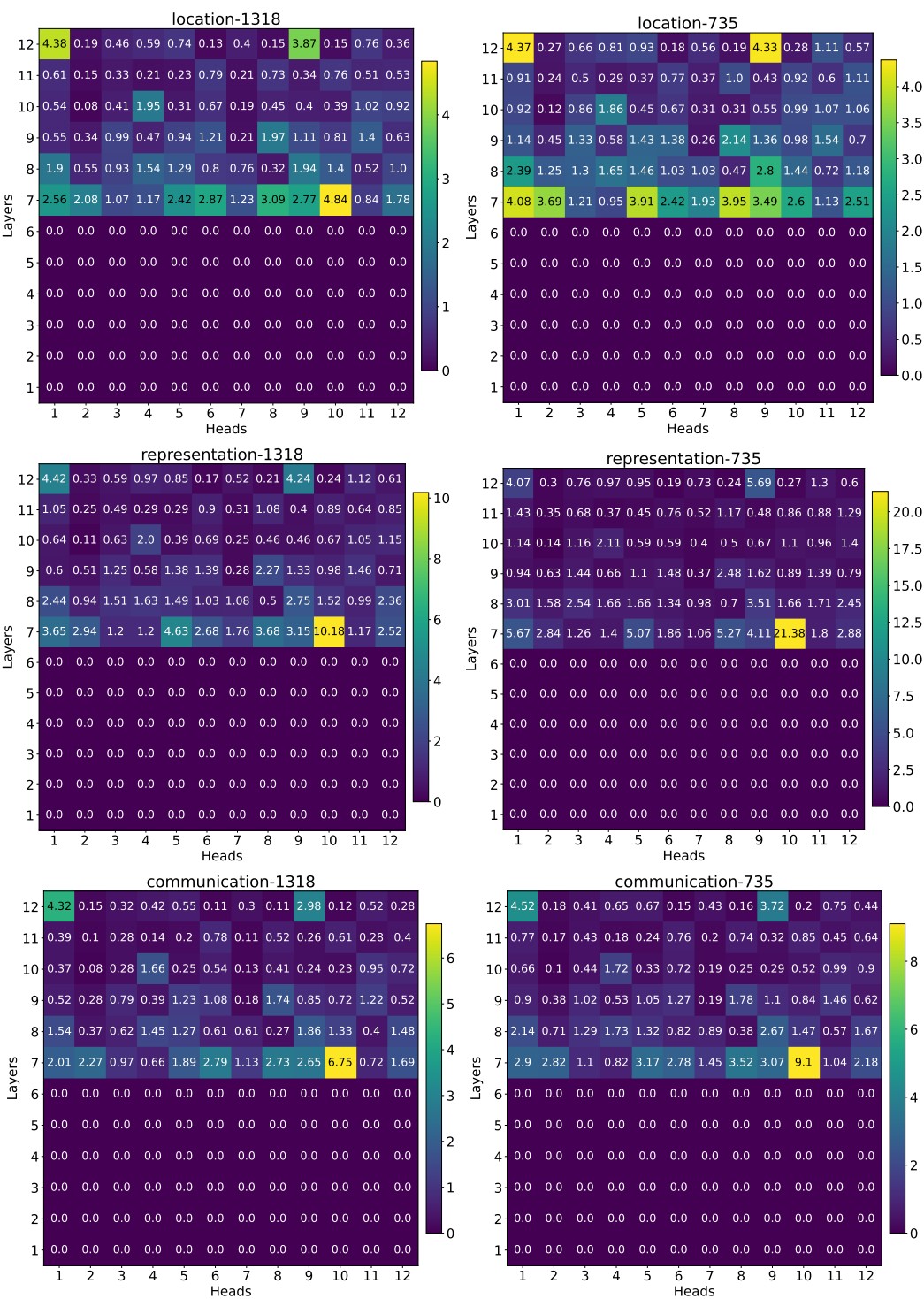

Figure 9: **Density percentage (%) of different heads across different attention layers.** Each row represents a different KG and each column is a different seed.

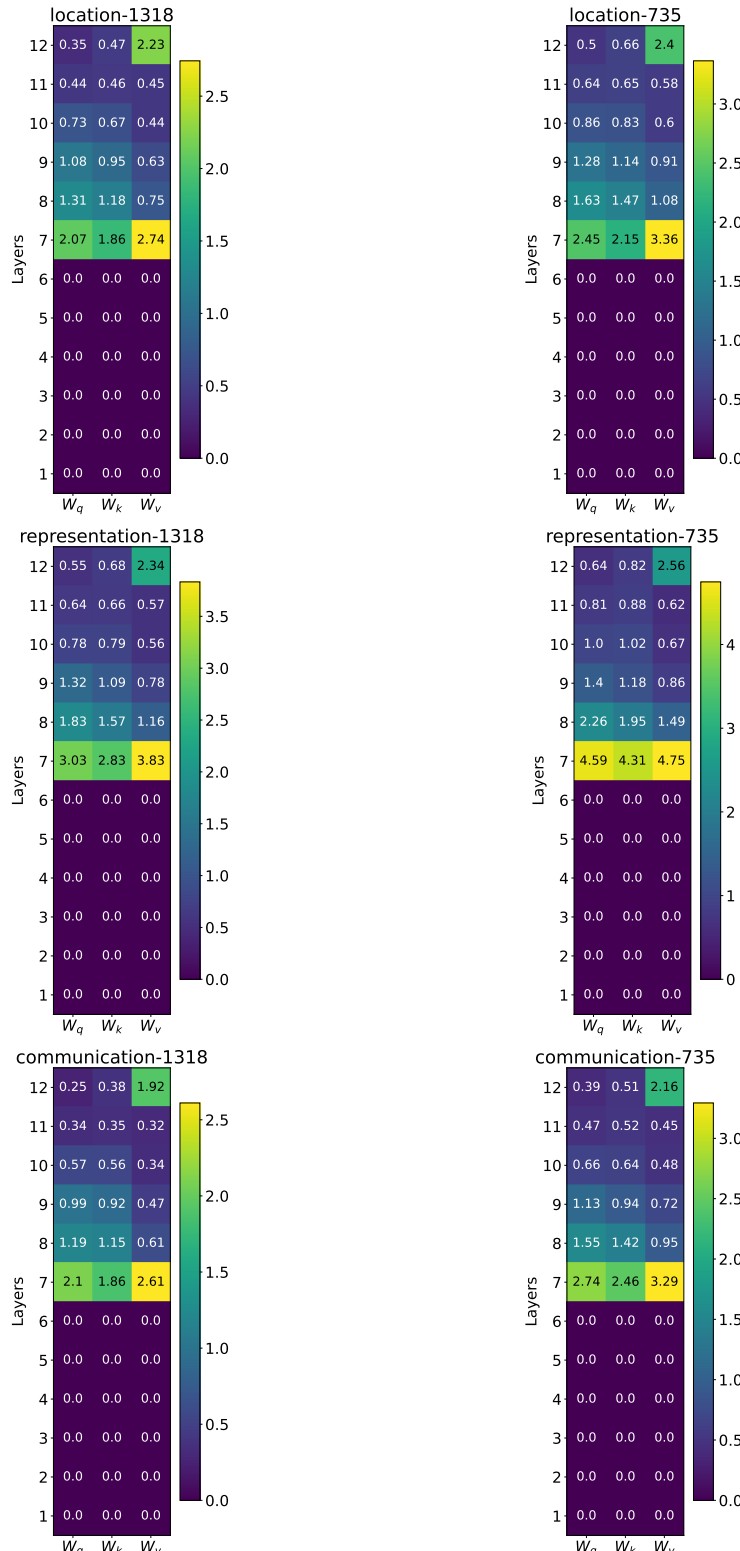

Figure 10: **Density percentage (%) of $W_q$, $W_k$, and $W_v$ masks in attention layers.** Each row represents a different KG and each column is a different seed.

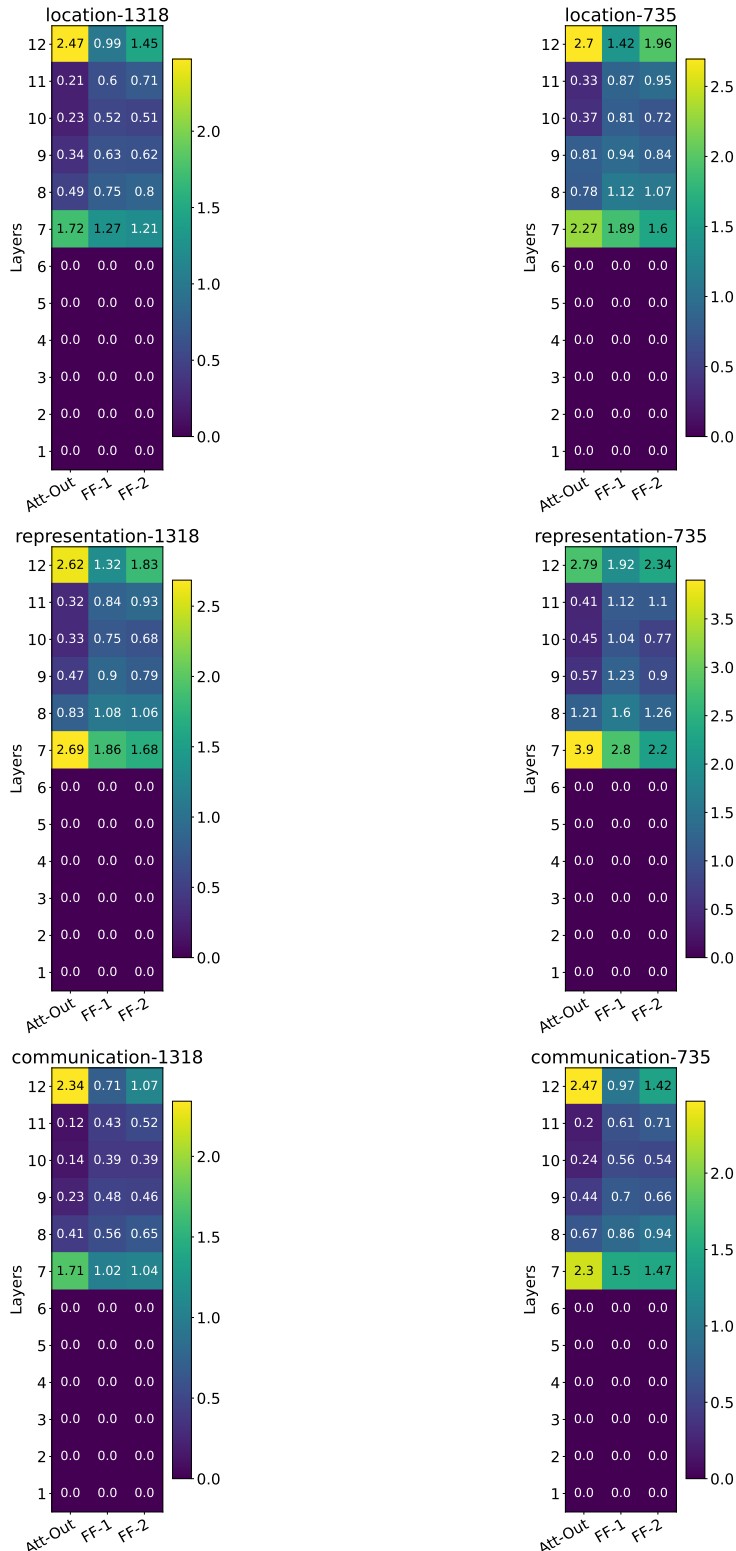

Figure 11: **Density percentage (%) of Att-Out, FF-1, and FF-2 masks.** Each row represents a different KG and each column is a different seed.

