# OpenReview forum: "Discovering Knowledge-Critical Subnetworks in Neural Language Models"
_ICLR.cc/2024/Conference — Submitted to ICLR 2024_

### Official Review · Reviewer_VC2F · 2023-10-27

**Soundness:** 2 fair
**Presentation:** 3 good
**Contribution:** 2 fair
**Rating:** 5
**Confidence:** 4

**Summary:**

The paper proposes to use a differentiable weight masking strategy to find subnetworks within pretrained language models that are critical for encoding specific knowledge. It is demonstrated that such subnetworks are highly sparse, and removing such subnetworks can selectively remove certain triplet knowledge without significantly affecting other knowledge and general language abilities in the model. The paper also shows that the knowledge-critical subnetworks determine the model's utilization of knowledge in downstream tasks.

**Strengths:**

* The authors introduce a novel method for finding subnetworks in large language models responsible for encoding specific knowledge, and demonstrate that it effectively finds highly-sparse subnetworks that are specific to a given set of knowledge.

* The paper uses analysis from different dimensions to verify the causal effect of the knowledge-critical subnetwork on the selected knowledge, including ablation study, expansion of the mask, and performance on downstream tasks controlled for the selected knowledge. The results strengthen the conclusion that the knowledge-critical subnetwork has a causal effect on the storage and expression of knowledge.

* The paper is well-written, well-structured and very accessible to the readers.

* The existence of knowledge-critical subnetworks may have significant implications on the interpretability of pretrained language models and could guide future research in the field. Future research may be able to explore how these subnetworks can be adapted, potentially leading to more efficient model fine-tuning.

**Weaknesses:**

* The effectiveness of knowledge removal is not quite clear due to limited metrics: the paper mainly uses "perplexity increase on verbalized triplet prompts" as a measure of knowledge removal but does not provide a solid interpretation of this metric. For example, one does not know how much perplexity increase corresponds to a complete (or near-complete) removal of the knowledge. Therefore, it may be hard to evaluate whether knowledge is truly removed with this metric alone. Also, there is a possibility that the perplexity drop is specific to the verbalizing template, so it may be helpful to evaluate on different templates as well.

  Perhaps knowledge-centric question-answering benchmarks, like those used in Section 6.4, could provide more interpretable results. However, the results currently presented in Section 6.4 also fail to prove reliable removal of knowledge, as pruning knowledge subnetworks only results in a small decrease in performance (3-4%).

* The effectiveness of knowledge preservation is also unclear due to possible overfit: the loss function is designed to preserve performance on ControlKG and ControlLM, and results show a negligible decrease in performance on them. However, it is possible that besides removing knowledge in TargetKG, the pruned model also sacrifices some other knowledge that is not in ControlKG and ControlLM. This would go against the goal of finding subnetworks specific to TargetKG. To rule out this possibility, the pruned model should be evaluated on a separate KG and a different corpus than those used in the loss function to make sure that the maintenance criterion is not overfitted to ControlKG and ControlLM.

* Lack of analysis on the discovered knowledge subnetworks

  * On sparsity: how sparse is truly sparse for a knowledge-critical subnetwork? If there are only 10-20 triplets in TargetKG, then 99% sparsity (~1M parameter for GPT2-small) does not seem very sparse, because it's unlikely that the model truly uses 1M parameter to store 10-20 triplets. I would personally expect a much higher sparsity (e.g., 99.99%) for a knowledge-critical subnetwork of 10-20 triplets, as a pre-trained language model typically stores a vast amount of knowledge.
  * On specificity: one way to examine the specificity of the subnetwork could be comparing the subnetwork for two different groups of triplets. If the subnetworks are largely different, it will provide evidence that the subnetworks are specific to the selected knowledge.
  * On trading-off suppression and maintenance: for two competing goals like suppression and maintenance, there is usually a tradeoff rather than a single best solution. It may be helpful to show this tradeoff by varying the weights in Equation 6, and it could also give justification for the choice of the weights.

**Questions:**

* On model choice: GPT2 is a slightly outdated model, recent models such as LLaMA are much better at knowledge tasks and may provide more statistically significant results.
* It's probably better to list some basic statistics of TargetKG, ControlKG, and ControlLM in the main text as they can be important for interpreting the results (e.g., sparsity).
* How does the expression criteria in Section 6.2 differ from the suppression criteria? It seems that Equation 7 is (approximately) just the reverse of Equation 3.

---

> ### Author Response · Authors · 2023-11-23
> **Response to Reviewer VC2F (Part 1)**
>
> We would like to thank reviewer VC2F for their helpful feedback. We are grateful that the reviewer finds our analysis vast and our paper well-written and accessible. The reviewer also stated that our work has significant implications for future work in model interpretability. We address their questions on *limited metrics* and whether *knowledge erasure is prompt dependent* in the general message to all reviewers. We wrote the requested changes and clarifications in the paper in purple.
>
> > The effectiveness of knowledge removal: Due to limited metrics, it is not clear whether knowledge removal is effective. Perplexity on verbalized triplet prompts alone is not enough.
>
> As mentioned in the general message to all reviewers, the reviewers wondered whether the rank of the gold tail entity can be used as an alternative metric. We also wondered about this and we report the rank differences between the remaining model and the original pretrained model in **Appendix D**, Table 10 (originally Table 11). These results align with the suppression and maintenance criteria described in Section 4.1. We added a forward pointer to these results in the main body in **Section 5** - “Success Metrics” paragraph.
>
> > The effectiveness of knowledge removal: Could the perplexity drop be specific to the verbalization template?
>
> We provide performance on paraphrases in **Appendix H** (originally Appendix G) for three WordNet hypernym TargetKGs. We have found that the average perplexity differences are high on paraphrases of TargetKG, and the perplexity differences for ControlKG either are near 0 or are negative. We added a forward pointer to these results in our main body in Section 6.3.
>
> > The effectiveness of knowledge preservation: Besides removing knowledge in TargetKG, the pruned model could sacrifice some other knowledge that is not in ControlKG and ControlLM. To make sure that the maintenance criterion is not overfitted to ControlKG and ControlLM the authors should use a separate corpus for ControlKG and ControlLM.
>
> As stated at the end of Section 5, paragraph “Datasets”, all results on ControlKG and ControlLM are on a held-out test set.
>
> > Lack of analysis on sparsity: 99% sparsity (~1M parameter for GPT2-small) for 10-20 triplets in TargetKG does not seem very sparse. I would personally expect a much higher sparsity (e.g., 99.99%) for a knowledge-critical subnetwork of 10-20 triplets, as a pre-trained language model typically stores a vast amount of knowledge.
>
> The reviewer has an interesting intuition on the size of knowledge-critical subnetworks, which we also share. An efficient and modular language model would ideally dedicate only a small amount of parameters to remove information on 10-20 triplets. However, it is not clear whether the amount of parameters needed to encode information scales with the amount of knowledge. In particular, the model could robustly encode information in a repeated manner across several parameters, since it has been originally pretrained with dropout. Moreover, this size could also be dependent on the knowledge type. For example, with limited analysis, we found that subnetworks for the representation KG tend to be larger than the location and communication KGs, although representation has fewer triplets than the rest. A hypothesis could be that more common or fundamental knowledge that affects other types of knowledge requires more parameters to be removed.
>
> We also want to emphasize that the sparsity was calculated over the masked regions rather than the whole subnetwork. Therefore, the majority of the subnetworks were a bit smaller than 1M parameters.

---

> > ### Author Response · Authors · 2023-11-23
> > **Response to Reviewer VC2F (Part 2)**
> >
> > > Lack of analysis on specificity: One way to examine the specificity of the subnetwork could be by comparing the subnetwork for two different groups of triplets. If the subnetworks are largely different, it will provide evidence that the subnetworks are specific to the selected knowledge.
> >
> > As mentioned in the general message to all reviewers, to address this inquiry, we created a new section in Appendix J called “Knowledge-Based Analysis”. We had similar findings as Appendix I, where the Intersection-over-Union was overall low (~3.56%).
> >
> > > Lack of analysis on trading-off suppression and maintenance: For two competing goals like suppression and maintenance, there is usually a tradeoff rather than a single best solution. It may be helpful to show this tradeoff by varying the weights in Equation 6, and it could also justify the choice of the weights.
> >
> > To address this concern, we run a minimal experiment on giving importance to one goal at a time for two KGs and one seed. Specifically when we set one of the $\lambda$ weights in Eq. 6 to a value of 3, we set the value of the rest to 1. We find that giving more weight to the Suppresion loss indeed finds checkpoints with higher perplexity differences on TargetKG while simultaneously satisfying the maintenance criteria. Giving more weight to the sparsity regularization on the other hand, ensures a high sparsity.
> >
> > This fits with the setting we use in all of our experiment, in which we give a larger weight to suppression ($\lambda_1=1.5$) and sparsity ($\lambda_4=3$) while the rest stay the same ($\lambda_{2,3}=1$).
> >
> > | $\lambda_1$ | $\lambda_2$ | $\lambda_3$ | $\lambda_4$ | Sparsity (%) | TargetKG $\Delta$ PPL | ControlKG $\Delta$ PPL | ControlLM $\Delta$ PPL | # of Valid Checkpoints |
> > |---|---|---|---|---|---|---|---|---|
> > | 1 | 1 | 1 | 3 | 99.5 [99.4, 99.5] | 350.2 [43.0, 657.4] | 0.1 [-0.1, 0.2] | 0.2 [0.2, 0.2] | 83.0 [7, 159] |
> > | 1 | 1 | 3 | 1 | 96.9 [95.6, 98.2] | 17.3 [-7.8, 42.4] | 1.0 [-2.0, 4.1] | 0.5 [0.2, 0.8] | 2.5 [0, 5] |
> > | 1 | 3 | 1 | 1 | 97.3 [96.4, 98.3] | 80.9 [59.2, 102.6] | -1.5 [-8.9, 5.8] | 1.0 [0.5, 1.5] | 53.5 [5, 102] |
> > | 3 | 1 | 1 | 1 | 98.5 [97.6, 99.3] | 12516.4 [682.2, 24350.6] | 0.0 [-1.4, 1.5] | 0.4 [0.2, 0.6] | 145.0 [133, 157] |
> >
> > > How do the expression criteria in Section 6.2 (now in Appendix G) differ from the suppression criteria? It seems that Equation 7 is (approximately) just the reverse of Equation 3.
> >
> > Eq. 3 and 7 are not approximately opposite equations. Eq. 3 suppresses the expression of the tail token given the head and relation tokens from the remaining model by optimizing for  a uniform prediction distribution. Eq. 7, on the other hand, minimizes the tail token cross-entropy loss on the removed subnetwork. The point we wanted to make with this experiment is that one objective does not guarantee the other, and in particular, the Eq 7. or in other words, the Expression loss, combined with the other objectives we proposed will lead to larger knowledge-critical subnetworks, which does not as fulfill the sparsity criterion.

---

### Official Review · Reviewer_njyU · 2023-10-31

**Soundness:** 3 good
**Presentation:** 3 good
**Contribution:** 2 fair
**Rating:** 3
**Confidence:** 3

**Summary:**

This paper investigates the presence of knowledge-critical subnetworks in pretrained language models (LMs) and proposes a method to discover and remove these subnetworks while minimizing adverse effects on the behavior of the original model.  Overall, the paper presents a novel approach for discovering knowledge-critical subnetworks in pretrained language models. However, further evaluation and comparison with existing methods, as well as addressing the mentioned weaknesses and clarifying the typos, would strengthen the paper's contribution.

**Strengths:**

1. Novel approach for discovering knowledge-critical subnetworks: The paper introduces a differentiable weight masking scheme that allows for the identification of subnetworks responsible for encoding specific knowledge in pretrained language models. This approach provides insights into how knowledge is encoded and can be potentially useful for model editing and finetuning.

2. Analysis of seed-based variance and subnetwork composition: The paper investigates the stability of subnetwork discovery under seed-based variance and explores the composition of subnetworks from different seeds. This analysis adds valuable insights into the robustness and generalizability of the proposed method.

**Weaknesses:**

1. Lack of comparison with other existing methods: The paper does not provide a comprehensive comparison with other existing methods for discovering knowledge-critical subnetworks in pretrained language models. This makes it difficult to assess the novelty and effectiveness of the proposed method.

2. Limited evaluation on downstream tasks: The paper primarily focuses on the discovery of knowledge-critical subnetworks but lacks a thorough evaluation of the impact of these subnetworks on downstream tasks. It would be beneficial to include experiments that demonstrate the effect of subnetwork removal on various NLP tasks.

**Questions:**

1. How does the proposed differentiable weight masking scheme compare to other existing methods for discovering knowledge-critical subnetworks in pretrained language models?

2. Can the discovered knowledge-critical subnetworks be used for targeted model editing or finetuning to improve specific task performance?

3. Can you provide more details about the filtering processes applied to the sampled connected KGs? How did these processes ensure the quality and balance of the sampled graphs?

4. How did you validate the effectiveness of the discovered subnetworks in suppressing the expression of target knowledge triplets?

---

> ### Author Response · Authors · 2023-11-23
> **Response to Reviewer njyU**
>
> We would like to thank reviewer njyU for their feedback. Notably, the reviewer found our approach as a novel way to discover knowledge-critical subnetworks. The reviewer considers our approach a potential future application in model editing and finetuning. We wrote the requested changes and clarifications in the paper in purple.
>
> > Can you provide more details about the filtering processes applied to the sampled connected KGs? How did these processes ensure the quality and balance of the sampled graphs?
>
> We describe the filtering processes applied to the sampled connected KGs in **Appendix A**. In summary, once a small connected KG is sampled, we apply two filtering processes. The first one enforces many-to-one relationships in the $K_T$ graph to avoid head entities with multiple tails. This ensures that the dataset is not dominated by certain head entities and balances the graph. The second filtering process reduces the tail-entity imbalance to avoid over-fitting to a small set of tokens. For this, we count the frequency of the tail tokens in the sampled graph and keep at most a quartile amount of triplets with shared tail entities.
>
> > How did you validate the effectiveness of the discovered subnetworks in suppressing the expression of target knowledge triplets?
>
> We validate the effectiveness of the discovered subnetwork by:
> 1. Measuring the effect on the TargetKG, a validation ControlKG, and a validation LModeling when the subnetwork is removed from the original model
> 2. Comparing with a randomly masked baseline to see if the increase in the perplexity can be matched with an equally sparse removed subnetwork
> 3. Doing an ablation study to test whether all of the terms are required to get the most effective suppressing effect
> 4. Verifying whether, for a downstream task, a forgotten test set knowledge can be transferred through finetuning – we find that removing the subnetwork indeed hinders this ability
> 5. Establishing whether worse scoring paraphrases of the triplet on average increase in perplexity for TargetKG and stay similar for ControlKG.

---

### Official Review · Reviewer_ikUx · 2023-11-01

**Soundness:** 1 poor
**Presentation:** 2 fair
**Contribution:** 2 fair
**Rating:** 3
**Confidence:** 4

**Summary:**

The paper argues that language models contain sparse subnetworks of parameters that are critical for expressing specific knowledge relationships. The authors then use an existing selected-pruning method to identify these "knowledge-critical" subnetworks: learning a binary mask over the parameters. They propose to optimise the binary mask jointly with three objectives 1) suppressing the expression of target knowledge triplets when the subnetwork is removed 2) maintaining performance on other knowledge triplets of a predefined KG and language modeling 3) sparsity regularizer as most work on pruning.

Experiments on GPT-2 variants find highly sparse subnetworks of  ~98% sparsity, which is similar as most sparsity work using hard concrete.  Interestingly, the authors show that, removing the subnetworks significantly reduce the model's ability to express the associated knowledge, but maintain other capacities, indicating a certain level of controllability over the specific knowledge.

**Strengths:**

- The authors address a very timely problem -- identify LLM subnetworks that correspond to certain knowledge and manipulate over the subnetworks to control the access of the knowledge.
- The authors propose a sensible approach for tackle the problem. Empirical experiments over GPT-2 show consistent trends in different datasets.

**Weaknesses:**

I think the paper can be improved in the following aspects:
- Lack of Meaningful Metrics.
    - The authors are showing the sparsity level over the masked parameters. Therefore 98% sparsity does not mean 98% of parameters in LLMs are not used. It would be more clear if the authors display the real sparsity levels.
    - The authors only use PPL to measure the effectiveness of their method. However, the ppl values can be a bit confusing as its range can be very big. Would it more sensible to use ranking metrics? for example, the rank of the target entity token?
- Lack of ablation studies
    - ablation on sparsity levels. does varying the sparsity level from 98% to 20% or 60% change the conclusion?
    - pruning methods based on hard concrete are usually sensitive to hyper-parameters. do you have any hyper-parameter sensitivity analysis?
- Limited baselines.
    - The only baseline is random masking of the maskable parameters, which is no surprise working poorly. The random baseline has no access to either TargetKG or ControlKG. It seems unfair to compare it with the proposed method if it uses much less information.

**Questions:**

- Does your method generalise to other pretrained language models?
- What's the computational complexity of the proposed method? Can it generalise to 7B parameter model [1]?
- What are the computational infrastructure for your experiments? If the readers want to reproduce your results, how many GPUs do they need?
- Prior research show that the embedding layer might contain lots of redundancy as the tokens follow long-tailed distribution. If you want to achieve high sparsity, pruning the embeddings can be a good choice. At the same time, the embeddings can be very informative. One simple baseline to remove a certain knowledge would be just erasing entity-related token embeddings. Why this is not one of your baseline?
- Does combining two knowledge-critical subnetworks lead to suppress of both pieces of knowledge?
- Does predicting missing entity fully represent this knowledge triplet? I am not sure. Even if it can correctly predict the missing entity, the prediction might be only based on the (subject, object) pair instead of based on the specific relation. In general, a knowledge triplet can be rephrased in multiple ways, eg. swapping the order of subject and object, missing relation prediction [2] etc. Can the proposed method can deal with the various rephrasing of a certain knowledge?

[1]: https://arxiv.org/abs/2302.13971

[2]: https://arxiv.org/abs/2110.02834

---

> ### Author Response · Authors · 2023-11-23
> **Response to Reviewer ikUx (Part 1)**
>
> We would like to thank reviewer ikUx for their thoughtful feedback. We appreciate that the authors find the problem very timely and our method as a sensible way to approach it. We also address the reviewer’s question on *limited metrics*, *cross-topic subnetwork analysis*, and *robustness to knowledge rephrasing* in the general message to all reviewers. We wrote the requested changes and clarifications in the paper in purple.
>
> > The authors are showing the sparsity level over the masked parameters. Therefore 98% sparsity does not mean 98% of parameters in LLMs are not used. It would be clearer if the authors displayed the real sparsity levels.
>
> Indeed, 98% sparsity means the sparsity of the removed subnetwork mask over the masked layers (in this case, the upper-half layers of the model). To make this more intuitive, we suggest reporting the **density of the removed subnetwork across all parameters** in the network, including those that are not masked. As it would affect the phrasing in many parts of the paper, we suggest making this revision for camera-ready.
>
> > Perplexity ranges can be very big. Would it be more sensible to use the rank of the target entity token?
>
> As mentioned in the general message to all reviewers, the reviewers wondered whether the rank of the gold tail entity can be used as an alternative metric. We also wondered about this, and so in the submission, we report the rank differences between the remaining model and the original pretrained model in **Appendix D**, Table 10 (originally Table 11). These results align with the suppression and maintenance criteria described in Section 4.1. We added a forward pointer to these results in the main body in **Section 5** - “Success Metrics” paragraph.
>
> We chose to display perplexity differences in the main body instead of rank as they better capture model certainty on the gold-tail token. For example, there could be cases where the remaining model has a larger rank than the original model leading to a large difference, but the probability differences between the two can be very small, and therefore not enough information about model certainty is reflected. Particularly, in our setting, where we optimize for a small difference between the TargetKG tail-token prediction distribution and a uniform distribution over all tokens in the vocabulary, we are bound to have multitudes of tokens being considered equally probable. This makes the rank metric less interpretable than perplexity. However, we agree that both metrics provide complementary views on this question, which is why we reported rank in the appendix initially.
>
> > Lack of ablation on sparsity levels – Does varying the sparsity level from 98% to 20% or 60% change the conclusion?
>
> The reviewer brings an interesting perspective from the point of view of compression. Differentiable-weight masking is popular as a compression technique (Sanh et al 2020, Zhao et al 2020). Varying the sparsity level would require implementing a different type of regularization which could be possible. However, we instead designed an aggressive sparsity regularization as our goal in finding a knowledge-critical subnetwork is to find the minimum compression ratio to achieve semantic suppression of the knowledge. As a result, while an interesting research direction on its own, we think this analysis may not align with the core inquiry of our paper.

---

> > ### Author Response · Authors · 2023-11-23
> > **Response to Reviewer ikUx (Part 2)**
> >
> > > Pruning methods based on hard concrete are sensitive to hyper-parameters. Do you have any hard-concrete hyperparameter sensitivity analysis?
> >
> > The reviewer is concerned about the sensitivity of hard-concrete hyperparameters. Past studies have shown the mask initialization hyperparameter tends to be sensitive (Zhao et al. 2020). We initially had run a small study to decide on the initialization. We re-ran this study with 2 KGs (location, communication) for the same seed (735) for 4 initializations (0.25, 0.45, 0.5, 0.75).
> >
> > We find that when the masking probability, or in other words, the probability of being included in the critical subnetwork is low – such as 0.25 – the hyperparameter often leads to worse maintenance on ControlKG and lower suppression of TargetKG.
> > On the other hand, a high initialization such as 0.75 can be a hit or miss for TargetKG suppression (as seen by the minimum and maximum values shown in the Table below), with overfitting on the maintenance datasets (i.e. decrease of perplexity in predicting ControlKG triplet tails compared to original model).
> > Between 0.5 and 0.45, where the latter is the initialization we used in our experiments, we don’t find a significant difference other than a slightly higher suppression effect for 0.45.
> >
> > Therefore our choice for mask initialization seems reasonable.
> >
> > | Initial Mask Prob | Sparsity (%) | TargetKG $\Delta$ PPL | ControlKG $\Delta$ PPL | ControlLM $\Delta$ PPL |
> > |---|---|---|---|---|
> > | 0.25 | 99.5 [99.5, 99.6] | 332.1 [119.5, 544.6] | 3.7 [2.6, 4.7] | 0.1 [0.1, 0.1] |
> > | 0.45 | 99.5 [99.5, 99.5] | 1287.8 [80.0, 2495.6] | 0.9 [-1.1, 2.9] | 0.2 [0.1, 0.2] |
> > | 0.5 | 99.4 [99.4, 99.5] | 939.1 [59.9, 1818.2] | -0.2 [-0.2, -0.1] | 0.2 [0.2, 0.2] |
> > | 0.75 | 98.9 [98.8, 99.1] | 13043.1 [4.6, 26081.5] | -1.9 [-1.9, -1.8] | 0.3 [0.3, 0.4] |
> >
> > > What is the computational infrastructure for your experiments? If the readers want to reproduce your results, how many GPUs do they need?
> >
> > All details on the computational infrastructure are in **Appendix B**, in particular in the “Hyperparameters” paragraph, where we define how many steps are required for how many A100 40GB GPUs. For all experiments, one can use a single A100 40GB GPU. However, we chose to speed up the process for the larger models (GPT2-medium and GPT2-large) by using data parallelism across 3 GPUs.
> >
> > > Does combining two knowledge-critical subnetworks lead to suppressing both pieces of knowledge?
> >
> > As mentioned in the general message to all reviewers, to address this inquiry, we created a new section in **Appendix J** called “Knowledge-Based Analysis”. We calculated the Intersection-over-Union of the subnetworks across different layers and layer types for the same seed but different KGs (Fig. 7) and composed subnetworks for the same seed and different KGs (Table 18).
> >
> > Using the union of the subnetworks increased the suppression effect across all the TargetKGs on average, but it also reduced the maintenance effect. Therefore, it may be possible to naively combine subnetworks across seeds or KGs, however, they may not guarantee the maintenance criteria to the same extent.
> >
> > > Can the proposed method deal with the various rephrasing of certain knowledge?
> >
> > We provide performance on paraphrases in **Appendix H** (originally Appendix G) for WordNet three hypernym TargetKGs. We have found that the average perplexity differences are high on paraphrases of TargetKG, and the perplexity differences for ControlKG either are near 0 or are negative. To clarify the takeaway, we have rephrased it and the changes can be seen in purple. We also added a forward pointer to these results in our main body in Section 6.3, as it is also a robustness analysis.

---

> > > ### Comment · Area_Chair_eDW2 · 2023-12-03
> > > **comment from reviewer  ikUx**
> > >
> > > Since the system is not allowing reviewer ikUx to make their comments visible to the authors, I post their comment here:
> > >
> > > I confirm I have read the authors' reply. The rebuttal does not change my opinion of the paper.

---

### Official Review · Reviewer_cTTe · 2023-11-02

**Soundness:** 2 fair
**Presentation:** 3 good
**Contribution:** 2 fair
**Rating:** 5
**Confidence:** 4

**Summary:**

This paper explores a method for detecting and zeroing-out the parameters of an LLM which contain knowledge relevant to a specific topic. Such masks are posited as forming sub-networks critical to the topic area. To do this, the authors formulate a loss function whose optimization has four objectives, each corresponding to a term in the loss function:

 1. topic-specific knowledge erasure: increase the perplexity on knowledge-base triples for the target topic area
 2. preserve knowledge about non-target topics: perplexity on knowledge-base triples from non-target topics should be maintained at baseline (pre-intervention) levels
 3. preserve general language fluency: the perplexity of non-target topic text should be maintained at baseline (pre-intervention) levels
 4. sparsity: the learned parameter mask should mask out few parameters of the original LLM -- i.e., it should be sparse

Experiments performed on triples from two different knowledge bases (WordNet and ConceptNet) show that the knowledge erasure procedure given does in fact increase perplexity on target-topic triples but not on triples from other topics. And the experiments show that basic linguistic fluency is maintained such that language perplexity does not increase for text outside of the target topic area.

The authors also perform experiments to examine whether knowledge erasure causes the effected LLMs to perform worse on downstream Q&A tasks for questions related the target topic. And indeed they do.

Finally, ablation studies are performed to determine the importance of the first three component terms of loss function; additional experiments are performed to determine whether the detected knowledge-relevant subnetwork masks are robust such that expanding or shrinking their area does not drastically alter the level of knowledge erasured; and a slightly different objective function based on "knowledge expression" is tried but abandoned for the objective above.

**Strengths:**

The question of how topic-specific knowledge and expertise is stored in LLMs is important and fascinating since very little is currently well understood; and it is practically useful for purposes like correcting factual errors and censoring bigotry learned by a language model. So I view this area of research as one that will be active for the years to come.

And the approach of finding topic relevant parameter networks by learning to erase subject-specific knowledge while preserving other knowledge appears rather novel and clever.

Last, the experimental results demonstrate that the objectives optimized in the loss are achieved.

**Weaknesses:**

While an interesting start, I feel as though the paper falls short of its promise. The big piece that feels missing from this paper is an analysis of the detected subnetworks. The experiments showed that the subnetworks typically consisted of between 1% and 2% of the network parameters. But which groups of parameters were they? And at which network layers? How distributed throughout the network were they? Did they consist of adjacent/localized blocks of parameters, or were they isolated and distributed? Was there any other sort of topological structure associated with these subnetworks?  And how might the masked regions be working to zero-out knowledge? These questions are not explored. (Moreover, in the appendix, the authors show that rerunning the knowledge erasure procedure from different random seeds finds alternative subnetworks that erase the same set of facts but that have with fairly little overlap with each other. This shows that the found critical subnetworks are not unique, so there is perhaps a bigger story to be told.)

Another question I felt wasn't sufficiently explored relates to the permanence of the knowledge erasure process. In particular, could a motivated individual recover the erased knowledge by using clever prompting or a fine-tuning process? That is to say, is the erased knowledge still in the network somewhere?

Last, I found section 6.3's use of the term "overfitting" to be a bit confusing, since overfitting means that the loss on the training set is significantly less on the test set. But that's not what's being examined here. Instead, it seems like section 6.3 is devoted to sensitivity analysis, to see how robust the perplexity differences are to changes in the mask. But I found the region growing and shrinking approach used here to be unwarranted since, as mentioned above, no analysis is performed to determine whether the masks even have regional (contiguous) structure.

**Questions:**

My main question is whether there is some analysis performed which reveals the nature of the found masks. Are they contiguous / network-like structures, or something else?

---

> ### Author Response · Authors · 2023-11-23
> **Response to Reviewer cTTe (Part 1)**
>
> We would like to thank reviewer cTTe for their insightful feedback. We thank the reviewer for emphasizing that very little is known about how pretrained language models represent knowledge. We also value their acknowledgement of the possible future applicability of our work in factual error correction. In addition to this comment, we also address the reviewer’s questions on *cross-topic analysis of subnetwork structures* and *whether knowledge erasure is prompt-dependent* in the general message to all reviewers. We wrote the requested changes and clarifications in the paper in purple.
>
> > The experiments showed that the subnetworks typically consisted of between 1% and 2% of the network parameters. But which groups of parameters were they?
>
> To address the question, we created new sections **Section 6.2** and **Appendix K** called “Structural Analysis”. While a more detailed analysis can be found in the revised paper, we summarize here our findings across three WordNet TargetKGs and three seeds:
> - Depth-wise: We have found that consistently the subnetwork is most dense around the first and final masked layers (in the case of GPT2-small, these would be layers 7 and 12).
> - Layer-type wise:
>     - For layer 7 and layer 12, we found that knowledge-critical subnetworks are most dense in the output layer of attention modules.
>     - However, in middle layers, the most dense layer types are the Feed-Forward networks, in particular, the first linear layer in each Feed-Forward network.
> - Contiguity/Locality/Distributedness:
>     - We have not found any complete columns or rows of weights that were dense in the critical subnetwork. This means that there are no input or output neuron features that get completely removed when the critical subnetwork is removed. Therefore the masked region may not be working to zero-out the knowledge by turning specific features off, which would counter the prevailing view that neuron-level changes are necessary for mechanistic interventions (Dai et al. 2022, Meng et al. 2022).
>     - However, in Figure 9, in the case of three WordNet TargetKGs and three seeds, regardless of the KG and seed configuration, we found that the 10th head in layer 7, and the 1st and 9th head in layer 12 are significantly less sparse. Therefore while the subnetworks are not overlapping according to the Intersection-over-Union metric, they tend to be dense in similar areas.
>
> > Was there any other topical structure associated with these subnetworks?
>
> As mentioned in the general message to all reviewers, to address this inquiry, we created a new section in **Appendix J** called “Knowledge-Based Analysis”. We did the same experiments as Appendix I (originally H), where we calculated the Intersection-over-Union of the subnetworks across different layers and layer types for the same seed but different KGs (Fig. 7) and composed subnetworks for the same seed and different KGs (Table 18).
>
> We had similar findings as **Appendix I**, where the Intersection-over-Union was overall low but most significant at the attention sublayer of the final transformer block. We also observed similar results for composing subnetworks. Therefore, it may be possible to combine subnetworks across seeds or KGs naively, however, they may not guarantee the maintenance criteria to the same extent.
>
> > Could a motivated individual recover the erased knowledge using clever prompting or a fine-tuning process? (i.e. the knowledge is still in the network)?
>
> We provide performance on paraphrases in **Appendix H (originally Appendix G)** for three WordNet hypernym TargetKGs. We have found that the average perplexity differences are high on paraphrases of TargetKG, and the perplexity differences for ControlKG either are near 0 or are negative. We added a forward pointer to these results in our main body in Section 6.3, as it is also a robustness analysis.
>
> Moreover, we show in **Section 6.4** that removing a knowledge-critical subnetwork harms the ability to transfer the knowledge to a downstream task. We have changed Tables 5 and 15 to clearly show accuracy differences from the original model (rather than the actual accuracy result for each baseline).
>
> If there were many explicit instances of the knowledge in the training set used to finetune the model, it’s possible that fine-tuning the model after removing the knowledge-critical subnetwork could re-form a representation to recover the knowledge. However, we also note that in the case of blackbox systems, users often have no access to finetune the underlying model, so removing subnetworks may represent a promising method for dampening the model’s knowledge of particular types of sequences.

---

> > ### Author Response · Authors · 2023-11-23
> > **Response to Reviewer cTTe (Part 2)**
> >
> > > In Section 6.3, the term "overfitting" can be confusing. Could it be changed?
> >
> > The reviewer mentioned that the term overfitting may be confusing in **Section 6.3** as overfitting is typically used when the loss on the training set is significantly less on the test set. To address this concern, we propose to reframe this section as “Are discovered subnetworks spurious suppression solutions or are they robust to perturbations?”. We highlight the changes made in purple. Please let us know if you prefer the “spurious subnetworks” perspective rather than “overfit subnetworks”.
> >
> > > In Section 6.3, if the subnetworks are not contiguous, then is growing/shrinking the model to evaluate robustness justified?
> >
> > The reviewer found the perturbation analysis unjustified because “no analysis is performed to determine whether the masks have contiguous structure”. However, as mentioned in the paper, we do not grow and shrink the subnetwork contiguously. We evaluate its robustness in suppressing the TargetKG and maintaining its modeling abilities by randomly removing/adding parameters from/to the remaining model. We believe that randomly removing further parameters from the remaining model, which is removing a superset of the knowledge-critical subnetwork, could reveal the spuriousness if TargetKG $\Delta$ PPL were to drop drastically.

---

### Author Response · Authors · 2023-11-23
**General Response**

We thank all of the reviewers for their thoughtful and constructive feedback. We wanted to answer and highlight some of the shared inquiries among the reviewers. We wrote the requested changes and clarifications in the paper in purple.

> Perplexity differences may not be enough and can get too big. Could we instead use a different metric (e.g. rank)?

The reviewers are concerned that perplexity differences may not be sufficient to measure knowledge suppression and suggest in particular the rank of the gold tail entity relative to other output tokens. We also wondered about this, and so in the submission, we report the rank differences between the remaining model and the original pretrained model in **Appendix D, Table 10 (originally Table 11)**. We observe that the target knowledge tail entity tokens are on average demoted in the prediction distribution. Moreover, the maintained knowledge rank differences are near zero. These results align with the suppression and maintenance criteria described in **Section 4.1**. We added a forward pointer to these results in the main body in **Section 5** - “Success Metrics” paragraph.

> Are there any topic-specific shared structures in the found knowledge-critical subnetworks? Moreover, could we compose these masks to simultaneously suppress knowledge from several critical subnetworks?

Several reviewers were intrigued as to whether knowledge-critical subnetworks of different KGs shared similar structures and whether they could be composed to suppress the union of their KGs. Our original submission explored a different axis in **Appendix I (originally H)**: the shared structures of subnetworks across seeds. We now add a second appendix, **Appendix J**, to address also the composability of these masks across different KGs for the same random seed.

As a brief reminder,  we presented seed-based variance and composition experiments in **Appendix I (originally H)**. We analyzed whether the subnetworks across different random seeds for the same KG shared structures and whether they could be composed. We concluded that the subnetworks shared very few exact parameters (i.e., low Intersection-over-Union) and that while using the union of the subnetworks increased the suppression effect on the TargetKG, the union also reduced the maintenance effect. Specifically, the control criteria could no longer be satisfied (with a PPL increase of ~30), but the target knowledge remained suppressed (with an even greater PPL increase of ~4000). The reviewers found this experiment interesting and wondered whether there could be a subnetwork overlap across KGs.

To address whether cross-knowledge subnetworks could be composable, we created a new section in **Appendix J** called “Knowledge-Based Analysis”. We did the same experiments as **Appendix I**, where we calculated the Intersection-over-Union of the subnetworks across different layers and layer types but instead for the same seed and different KGs (Fig. 7) and composed subnetworks for the same seed and different KGs (Table 18). We observed that low Intersection-over-Union across masks for different target KGs. We also observed similar results for composing subnetworks: using the union of the subnetworks increased the suppression effect across all the TargetKGs on average (PPL increase ~2000), but it also slightly reduced the maintenance effect (with a PPL increase of ~45 on the ControlKG). Therefore, it may be possible to naively combine subnetworks across seeds or KGs, however, the maintenance criteria may suffer. A future idea could be to continue optimizing for the subnetwork mask by initializing it as the union of the subnetworks to see if more robust suppression can be achieved.

> Is the method robust to a variety of paraphrases?

We provide performance on paraphrases in **Appendix H (originally Appendix G)** for three WordNet hypernym TargetKGs. We found that even when the knowledge is paraphrased, the perplexity increases for queries about the TargetKG (satisfying suppression) and remains equal or even slightly decreases (satisfying maintenance) on the ControlKG. To clarify the takeaway, we have rephrased it and the changes can be seen in purple. We also added a forward pointer to these results in our main body in **Section 6.3**, as it is also a robustness analysis.

> Is the method generalizable to other pretrained models?

Due to the limited rebuttal period, we cannot accomplish experiments on pretrained models other than GPT2. However, we plan on including them in the final version of the paper. In particular, all of the reviewers wanted to see an analysis with LLAMA 7B. We want to emphasize that the method can generalize to a variety of model sizes already, such as the  GPT2-large, which holds promise for LLAMA 7B.

---

### Meta-Review · Area_Chair_eDW2 · 2023-12-03

**Metareview:**

This paper introduces a method to identify subnetworks of a language model that are responsible for encoding specific knowledge, based on a multi-objective weight masking loss. When applied to various GPT2 models, the method uncovers subnetworks that encode specific kinds of ConceptNet/WordNet-like relational knowledge.

The reviewers and I agree that this paper tackles a timely and important challenge (understanding how language models work at a mechanistic level), and is original in its analysis. The reviewers however raised a number of methodological concerns, as well as concerns about what are the broader conclusions that we can draw from the study. Some of the major concerns were:

- lack of characterization of the discovered knowledge networks
- difficulty in interpreting the perplexity metric
- does the method generalize to other pre-trained models (especially larger ones), and would the results be confirmed?
- are there alternative, simpler ways to highlight knowledge sub-networks, and how would they compare to the computationally intensive procedure proposed by the authors?

The authors addressed the first two issues above and other concerns in the response and in a posted revision of the paper. Unfortunately, this led to a rather "messy" state, where many of the more critical results are in appendices.

I would personally be willing to trust the authors, given the amount of work they have already done to accommodate the reviewers' comments in the brief rebuttal time window, to incorporate the critical results into the main paper. However, I am not comfortable overturning the judgment of 4 reviewers, all agreeing that the paper is not mature enough to be published, even after the rebuttal.

**Justification For Why Not Higher Score:**

In its current state, the paper leaves to many important questions to the appendices, which greatly reduces its clarity and potential impact.

**Justification For Why Not Lower Score:**

N/A

---

### Decision · Program_Chairs · 2024-01-16

Reject